# To what extent does $CO_2$ diurnal cycle impact flux estimates derived from global and regional inversions?

Saqr Munassar[1,2,3], Christian Rödenbeck[1], Michał Gałkowski[1,4], Frank-Thomas Koch[5], Kai U. Totsche[2], Santiago Botía[1], Christoph Gerbig[1]

[1]Department of Biogeochemical Signals, Max-Planck Institute for Biogeochemistry, Jena, Germany
[2]Institute of Geoscience, Friedrich Schiller University, Jena, Germany
[3]Department of Physics, Faculty of Science, Ibb University, Ibb, Yemen
[4]AGH University of Kraków, Faculty of Physics and Applied Computer Science, Kraków, Poland
[5]Meteorological Observatory Hohenpeissenberg, Deutscher Wetterdienst, Hohenpeißenberg, Germany

*Correspondence to*: Saqr Munassar (smunas@bgc-jena.mpg.de)

**Abstract.** Ignoring the diurnal cycle in surface-to-atmosphere $CO_2$ fluxes leads to a systematic bias in $CO_2$ mole fraction simulations sampled at daytime, because the daily mean flux systematically misses the $CO_2$ uptake during the daytime hours. In an atmospheric inversion using daytime-selected $CO_2$ measurements at most continental sites and not resolving diurnal cycles in the flux, this leads to systematic biases in the estimates of the annual sources and sinks of atmospheric $CO_2$. This

study focuses on quantifying the impact of this diurnal cycle effect on the annual carbon fluxes estimated with the CarboScope (CS) atmospheric inversion at regional, continental, and global scales for the period of time 2010–2020. Our analysis is based on biogenic fluxes of hourly Net Ecosystem Exchange (NEE) obtained from the data-driven FLUXCOM-X estimates, together with global and regional atmospheric transport models. Differences between $CO_2$ mixing ratios simulated with daily averaged and hourly NEE from FLUXCOM-X range between around -2.5 and 7 ppm averaged annually throughout a site network across

the world. These differences lead to systematic biases in $CO_2$ flux estimates from the atmospheric inversions. Although the impact on the global total flux is negligible (around 2% of the overall land flux of -1.79 Pg C yr$^{-1}$), we find significant biases in the annual flux budgets at continental and regional scales. For Europe, the annual mean difference in the fluxes arising indirectly from the diurnal cycle of $CO_2$ through the boundary condition amounts to around 48% of the annual posterior fluxes (0.31 Pg C yr$^{-1}$) estimated with CarboScope-Regional (CSR). Furthermore, the differences in NEE estimates calculated with

CS increase the magnitude of the flux budgets for some regions such as northern American temperate and northern Africa by a factor of about 1.5. To the extent that FLUXCOM-X diurnal cycles are realistic at all latitudes and for the station set including many continental stations as used in our inversions here, we conclude that ignoring the diurnal variations in the land $CO_2$ flux leads to overestimation of both $CO_2$ sources in the tropical lands and $CO_2$ sinks in the temperate zones.

## 1 Introduction

Accurate estimation of $CO_2$ carbon budget is necessary for verifying the reduction of global carbon emissions in line with climate adaptation policies adopted in the Paris Climate Agreement (Glanemann et al., 2020). From a scientific perspective, it is also of high interest to extend our understanding about physical and biogeochemical dynamics of the carbon cycle in the

Earth's climate system. As reported in the Sixth Assessment Report (AR6) of the Intergovernmental Panel on Climate Change (Canadell et al., 2023), the cumulative carbon budget in the atmosphere has recently exceeded 279 Pg C by 2019 mainly due to fossil fuel combustion, cement production, and net emissions associated with land-use change since the industrial era began in 1850. The consequence is global warming with a global mean temperature excess by around 1.1°C between 1850-1900 and 2011-2020. Recently, the annual fossil-fuel related emission of $CO_2$ increased to 10.9 Pg C yr$^{-1}$ over the past decade (2010-2019), with 5.1 Pg C yr$^{-1}$ accumulated in the atmosphere, 3.4 Pg C yr$^{-1}$ taken up by the terrestrial biosphere, and 2.5 Pg C yr$^{-1}$ counted as an ocean sink (Canadell et al., 2023). Fixing the largest amount of atmospheric $CO_2$ in the biosphere via photosynthesis, the terrestrial biosphere sink consequently increases in response to $CO_2$ concentration growth, which is thought to result from three effects: 1) fertilisation by the rise of $CO_2$ abundance in the atmosphere, 2) increase of nitrogen inputs into soil used by agriculture to enhance plants growth, and 3) prolongation of growing seasons (Friedlingstein et al., 2022). It is important to note, however, that interannual variability in the land sink is large (about 1 Pg C yr$^{-1}$), making it difficult to pin down small emission variations in global atmospheric $CO_2$ concentrations (Baker et al., 2006). Moreover, due to the heterogeneity in terrestrial ecosystems, the atmosphere-to-land $CO_2$ fluxes are also largely variable at temporal and regional subscales (Marcolla et al., 2017), as regularly demonstrated by both Dynamic Global Vegetation Models (DGVMs) and data-based global inversion models, e.g. within the framework of the Global Carbon Project (Friedlingstein et al., 2022; Friedlingstein et al., 2023).

Atmospheric inverse modelling has increasingly been used at global (Ciais et al., 2010; Kaminski et al., 1999; Peylin et al., 2002; Rödenbeck et al., 2003) and regional scales (c.f. (Berchet et al., 2021; Chevallier et al., 2014; Gerbig et al., 2003; Monteil and Scholze, 2021; Munassar et al., 2022; Rivier et al., 2010)) to constrain the surface-atmosphere carbon fluxes and their variabilities based on atmospheric measurements sampled through surface in-situ networks and airborne-based monitoring instruments. Although the observational constraint is meant to drive the solution of the inverse problem, uncertainties and biases in atmospheric transport together with prescribed flux components such as anthropogenic emissions and ocean fluxes can diminish the capability of the inversions in terms of finding the optimal values of the state parameters (Engelen, 2002; Gurney et al., 2003). The atmospheric transport uncertainty especially remains a genuine challenge in inverse modelling to estimate carbon fluxes (Deng et al., 2017; Gerbig et al., 2008; Munassar et al., 2023), specifically over regions where observations either do not exist or are relatively sparse, such as over the tropics (Gurney et al., 2003; Botía, 2022). In fact, already in 2007 Stephens et al. (2007) have found inconsistencies in the vertical atmospheric $CO_2$ distributions between aircraft measurements and atmospheric models that transfer large amounts of terrestrial carbon from tropics towards northern latitudes, leading to overestimation of tropical sources and stronger sinks in northern terrestrial land, an effect necessary to maintain mass balance in the carbon budget.

The interhemispheric gradient of the annual mean $CO_2$ concentration evident by global atmospheric data represents one of the primary atmospheric constraints of the global carbon budget in all global atmospheric tracer inversions (Tans et al., 1990). Originally, this gradient was suggested to result primarily from higher emissions of fossil fuels over the Northern Hemisphere as compared to those in the Southern Hemisphere. Denning et al. (1995), however, found a significant meridional gradient

imposed by the seasonal $CO_2$ exchange of terrestrial biota, amounting to half of the gradient imposed by fossil fuel emissions, suggesting a stronger sink in the Northern Hemisphere than previously assumed. This effect has been referred to as "the atmospheric rectifier", in which $CO_2$ uptake and vertical mixing are both driven by solar radiation (Larson and Volkmer, 2008;

Denning et al., 1999). That is, in the early afternoon on any sunny day of the growing season, the uptake of $CO_2$ by photosynthesis and the dilution due to deep atmospheric vertical mixing attenuate the level of atmospheric $CO_2$ concentrations (Stephens et al., 2007). By contrast, during nighttime $CO_2$ concentrations accumulate near the Earth's surface owing to ecosystem respiration and the atmospheric stratification under shallow boundary layer. On daily average, the covariance between the atmospheric transport and terrestrial biospheric fluxes results in vertical gradient of $CO_2$ distributions with a

surplus near the surface. In addition,  an inter-hemispheric gradient is imposed with higher concentration near northern polar regions and lower in the tropics and Southern Hemisphere (Chan et al., 2008). This effect becomes uncertain at regional scales along the latitude bands. Therefore, misrepresentation of biosphere fluxes and vertical transport affects the simulations of $CO_2$ mole fractions (Gurney et al., 2003). Although the ability of transport models to represent the atmospheric variability is essential to reconstruct the observations of $CO_2$ mole fractions, Patra et al. (2008) found that the representation of $CO_2$

terrestrial input fluxes is crucial to capture $CO_2$ synoptical variations with best agreement with observations achieved when diurnally varying terrestrial fluxes are used. Increasing both the horizontal resolution and vertical levels in transport models improves the performance of transport models to retrieve $CO_2$ diurnal variations observed at different locations, especially over coastal and mountain terrains (Law et al., 2008).

As biases in simulated $CO_2$ mole fractions arise from both diurnal and seasonal variations of the vertical transport and biosphere

$CO_2$ fluxes, information of these variations should be accounted for in inverse modelling. While the seasonal effect can be reconciled in atmospheric inversions by the seasonal variations of $CO_2$, already included in the observational constraint, the diurnal variations are commonly missing in the observational constraint, because in inversions the use of observations is typically restricted to times when the lower atmosphere is well-mixed (i.e. during local noon to afternoon times) (Gerbig et al., 2008). An accurate representation of the such variations is then dependent on the performance of atmospheric transport models

to represent the seasonal and diurnal cycles. Therefore, reconciliation of the diurnal effect in atmospheric inversions is achievable by including diurnal variations of the biogenic terrestrial fluxes in the prior fluxes, provided that the vertical transport diurnal variations are accounted for. However, not all atmospheric inversions contributing to the Global Carbon Project (GCP) account for the diurnal cycle in biosphere-atmosphere exchange of $CO_2$, at least in their current setups such as CarboScope and the Carbon in Ocean-Land-Atmosphere inversion COLA (Liu et al., 2022). On the other hand, there are a

number of inversions using biosphere flux models that resolve diurnal variations, such as: the Copernicus Atmosphere Monitoring Service inversion CAMS (Chevallier et al., 2017) assimilating 3-hourly biosphere-atmosphere fluxes from ORCHIDEE (Krinner et al., 2005); the Global ObservatioN-based system for monitoring Greenhouse GAses inversion GONGGA (Jin et al., 2024) from ORCHIDEE-MICT (Guimberteau et al., 2018); the Global Carbon Assimilation System (GCAS) from the Boreal Ecosystems Productivity Simulator (BEPS) (Jiang et al., 2021); and both Carbon Tracker Europe

CTE (van der Laan-Luijkx et al., 2017) and the inversion algorithm THU (Kong et al., 2022) using 3-hourly NEE from the

Simple Biosphere model SiB4 (Haynes et al., 2019). Additionally, several GCP inversions utilise biogenic prior fluxes estimated by biosphere models without a direct inclusion of the $CO_2$ diurnal cycle (i.e., run at daily or coarser time steps). However, the diurnal cycle of $CO_2$ is included through temporal downscaling of gross flux components to hourly time steps using surface radiation and temperature from meteorological fields to derive the diurnal variations of GPP and ecosystem respiration following the approach established by Olsen and Randerson (2004). Of those inversion models are Carbon Tracker CT (Jacobson et al., 2023) , Carbon Monitoring System Flux CMS-Flux (Liu et al., 2021), Model for Interdisciplinary Research on Climate version 4 MIROC4-ACTM (Chandra et al., 2022), NICAM-based Inverse Simulation for Monitoring $CO_2$ (NISMON-$CO_2$) (Niwa et al., 2022), and the Institute of Atmospheric Physics Carbon dioxide retrieval Algorithm for satellite remote sensing IAPCAS (Yang et al., 2021).

Our study aims to investigate the impact of the diurnal cycle on global and regional $CO_2$ flux estimates using CarboScope inversion (CS), which so far (version v2023) does not account for diurnal variations in the $CO_2$ flux. In the standard global CS framework, the posterior biogenic fluxes are dominantly driven by the atmospheric data constraint, and the control vector represents deviations from zero biogenic fluxes used as a priori estimate. Unfortunately, the diurnal flux variability cannot be constrained by the atmospheric data, because the atmospheric measurements are only used during day-time hours for surface stations and night-time hours for mountain stations, reflecting that the transport model is expected to have particularly large errors outside these time periods. In the global total carbon flux, the impact of the diurnal cycle effect is attenuated because it is well constrained by the linear rise of $CO_2$ from year to year, due to the mass conservation of $CO_2$ in the atmosphere. However, for the fluxes at local scales the diurnal cycle effect can lead to biases in the estimates.

In studies focusing on specific land regions, like Europe, it is essential to make use of as many continental measurements as possible. As these measurements are typically located within complex atmospheric circulation patterns, the relatively coarse global transport model is not well suited to represent these measurement locations. Therefore, variations in transport and in fluxes need to be resolved on finer spatial and temporal scales. Unfortunately, due to computational limitations, the transport model, and thus the inversion calculation, needs to be confined to the regional domain of interest then. As the inversion problem is intrinsically global, however, such a regional inversion needs to be nested into a global inversion. Here, we consider the regional inversion for Europe using CarboScope-Regional (CSR) as described in (Munassar et al., 2023; Munassar et al., 2022), which is using the two-step scheme (Rödenbeck et al., 2009) to provide boundary conditions from a global inversion to the regional inversion. Even though the set-up of the regional inversion does include the diurnal cycle in the a priori fluxes, it is nevertheless prone to biases passed on through the lateral boundary conditions calculated by the global inversion currently not taking the diurnal cycle of $CO_2$ fluxes into account. In addition, the inversion is affected by imperfect representation of diurnal cycle in the transport model.

In this study, we focus on the impact of not having the diurnal cycle in the current global CarboScope set-up, both on the result of the global inversion and the inherited impact on the CSR results for Europe. Differences in $CO_2$ mole fractions simulated with and without diurnal variations in NEE are calculated by forward runs of the global transport model. These differences are then inverted by a global inversion to address the impact in flux estimates. We analysed the flux differences on the global

scale, in coarse latitude bands, as well as over a set of regions covering the whole globe. After that, the indirect effect of diurnal variations passed to regional inversions via the far-field influences was evaluated by using CSR in the regional domain of Europe employing a mesoscale transport model.

The organisation of the manuscript is as follows: the next section describes the methods applied in this experiment elucidating the inversion setup, the prior information obtained from the biosphere flux model, and the atmospheric transport models (Section 2). Results of $CO_2$ mixing ratio differences calculated with biosphere fluxes through forward model simulations across site network are presented in Section 3, as well as differences in NEE estimates due to the impact of the diurnal cycle effect. Section 4 is devoted to the discussion of the results analysed at global and regional scales highlighting the implications for carbon budget estimates. We summarise the key findings and relating perspectives of this study in Section 5.

## 2 Methods

### 2.1 Simulating the diurnal rectification of atmospheric $CO_2$

Modelled $CO_2$ mixing ratios are calculated for an atmospheric station network distributed around the world, as typically used for a global atmospheric inversion (Fig. 1). The CS global inversion uses atmospheric data collected from 169 stations around the world. Part of the stations provide continuous measurements, typically at hourly time intervals, while others provide flask samples collected at discrete times (typically weekly). According to the design of this experiment, we chose a subset containing 78 stations that consistently provide data from 2010 onwards, as in the CarboScope inversion run s10oc_v2022 (http://www.bgc-jena.mpg.de/CarboScope/?ID=s10oc_v2022). The transport model TM3 (Heimann and Körner, 2003) is run with biogenic terrestrial fluxes from FLUXCOM-X (Bodesheim et al., 2018; Jung et al., 2019; Nelson et al., 2024). As land NEE has the largest diurnal cycle, only the biogenic flux component is taken into account to simulate $CO_2$ dry mole fractions. Two forward runs are performed to generate simulations of $CO_2$ dry mole factions: one with diurnal variations in NEE and one without diurnal variations (by using daily averaged fluxes). Afterwards, the differences between simulations based on daily averaged NEE and simulations based on hourly NEE were calculated for each monitoring site across all the network. These differences in simulated $CO_2$ dry mole fractions are inverted in CS to quantify the impact in NEE estimates. This was done for practical considerations, as inverting a mole fraction difference is equivalent to performing two inversions and then obtaining the difference between the retrieved fluxes, due to the linearity of atmospheric tracer transport and inverse estimation.

### 2.2 Inversion setups

CarboScope (CS) is used at the global scale using the transport model TM3 (Heimann & Körner, 2003) at a spatial resolution of 5°x4°. As the standard CarboScope $CO_2$ inversion uses fixed ocean $CO_2$ fluxes from an interpolation of surface-ocean $pCO_2$ data, the state space vector is confined to the biogenic terrestrial flux component that is corrected spatially and temporally based on Bayesian inference (Enting, 2002). As the fluxes are resolved on a daily time step, the diurnal cycle in terrestrial ecosystems is not accounted for. The spatial and temporal autocorrelations of the prior error are exponentially decaying

functions with 1200 km spatial correlation length and 30 days temporal correlation length. The inversion algorithm searches for the optimal flux adjustment as an additive correction based on the constraints guided by observations and a-priori fluxes. For more details of the mathematical setup of CS, the reader is referred to Rödenbeck (2005).

To investigate the effect of the $CO_2$ diurnal cycle on regional fluxes, CSR is used to optimise NEE at 0.5°x0.5° spatial and 3-hourly temporal resolutions over specific regional domains, such as Europe in this study. A spatial correlation length of 66 km is defined for prior flux error in CSR covariance matrices, while the temporal error structure remains identical to the configuration of CS. The spatial correlation decay follows a hyperbolic function with the decay being faster in the meridional direction than in the zonal direction by a factor of 2. By comparison with CS, the TM3 transport model is replaced with the regional atmospheric transport model STILT with 0.25° horizontal resolution. It is important to note that a-priori biogenic fluxes used in CSR do account for the diurnal cycle. Thus, our investigation focuses on the influence of the $CO_2$ diurnal cycle as passed into the regional domain through initial and boundary conditions. The lateral boundary conditions are provided to the regional inversion by the two-step inversion scheme explained in Rödenbeck et al. (2009). Details of CS and CSR configurations, including prior uncertainty prescription, are listed in Table 1. For this experiment setup, ocean fluxes and anthropogenic emissions are omitted in both CS and CSR, because these cancel out in the difference.

The uncertainty of the model-data mismatch is defined similarly in CS and CSR. It comprises a combination of the uncertainties arising from measurements, atmospheric transport, and spatial representation. Weekly values of the errors are assigned to stations based on a classification regarding the ability of atmospheric transport model to represent atmospheric dynamics over the locations of stations. For instance, tall towers, mountain sites, and stations located at/near shores and aircraft samples have an error of 1.5 ppm, while surface sites that represent complex circulations are assigned with a relatively larger error (2.5 ppm). For hourly measurements, the error value is inflated depending on the number of data points assimilated per week such that the hourly error becomes the weekly error times the square root of the number of weekly hours (e.g., 42 hours in case a time window of 6 hours per day is chosen and if there are no data gaps).

## 2.3 Transport models

TM3 is an Eulerian transport model that solves the continuity equation (plus parametrisations of boundary layer and convective mixing) for atmospheric tracers in a three dimensional grid over the globe (Heimann and Körner, 2003). The model is driven by meteorological fields, such as wind velocity, air temperature, surface pressure, and specific humidity, obtained from NCEP reanalysis data (Kalnay et al., 1996). The tracer advection is determined by the mixing ratio and gradient of the tracer in grid-boxes based on the slopes scheme developed by Russell and Lerner (1981). In addition, the vertical transport is resolved by vertical diffusion and cumulus cloud transport deduced through evaporation fluxes, which are taken from meteorological fields. TM3 is run here at 5°x4° spatial resolution with 19 vertical levels spanning the troposphere and the stratosphere. Since the model is initialised with a homogeneous background of the tracer concentration, running the model for at least one year before the period of interest is done to avoid any impact resulting from the model spin-up.

In the regional inversion CSR, the Stochastic Time-Inverted Lagrangian Transport model STILT (Lin et al., 2003) is used at a finer horizontal resolution of 0.25°x0.25° to resolve the atmospheric mesoscale variability via tracking the dispersion of tracers backward in time from starting locations "receptors". Forecasted meteorological data obtained from the ECMWF Integrated Forecasting System (IFS) drive an ensemble of virtual particles at receptor locations, normally over stations where atmospheric data are sampled. The particles are transported ten days backward and the surface influence functions ("footprints") are stored at 0.25° horizontal resolution and hourly time steps. A daytime window of six hours (11:00 to 16:00 LT) is chosen for low elevation stations. For high altitude sites such as mountain stations, a nighttime window of 23:00 to 04:00 LT is used to select free troposphere conditions. Note that the vertical resolution of the underlying meteorological data is much higher for STILT (89 levels up to about 20 km height) compared to TM3.

## 2.4 Biogenic terrestrial fluxes

In Bayesian inversion formalism, an a-priori knowledge of the control parameters is essential to regularise the solution of the underdetermined system (Enting, 2002). Typically, prior fluxes of $CO_2$ representing the exchange between the surface and the atmosphere are taken from bottom-up estimation. As we aim to assess the impact of $CO_2$ diurnal cycle on the flux estimate, the fluxes are confined to the biogenic terrestrial component. The experiment was designed to invert the differences in mole fractions simulated with two variants of biosphere fluxes (i.e., with and without the diurnal cycle in the biosphere fluxes), rather than the formal way of implementing two inversions that would account for such variations in the prior biosphere fluxes. Hourly NEE calculated from FLUXCOM-X (Nelson et al., 2024), a data-driven upscaled flux product developed based on FLUXCOM (Bodesheim et al., 2018), is used as a variant that includes information of the $CO_2$ diurnal cycle in CS. In addition, another variant of these fluxes was created by averaging these fluxes from hourly to daily. Moreover, the dataset used in this study are produced at 0.05° spatial resolution as a new product (labelled as X-BASE) of the new modelling framework FLUXCOM-X, which has been developed to allow for applying different methodological choices with machine learning-based gradient regression algorithms (Nelson et al., 2024). FLUXCOM-X mainly targets the estimations of four predicted variables (NEE, Gross Primary Productivity (GPP), Evapotranspiration (ET), and Transpiration ($ET_T$)) using twelve predictor variables obtained from global meteorology and satellite observations of daily surface reflectance and land surface temperature from the MODerate Imaging Spectroradiometer (MODIS). Hourly ERA-5 reanalysis data including several parameters such as air temperature, incoming shortwave radiation, and vapour pressure deficit are used at 0.25° spatial resolution. A number of 294 eddy covariance stations distributed across the world was used to provide observed fluxes for training and cross-validation. Quite a good performance of FLUXCOM-X was exceptionally noticed in retrieving the diurnal variations of the predicted variables, evident by comparing the model predictions with observations withheld for validations throughout the globe. This promotes the validity of using such products in our analysis to quantify the impact of $CO_2$ diurnal cycle in inverse modelling. For detailed information about X-BASE products and setup, the redear is referred to Nelson et al. (2024).

To quantify the impact of $CO_2$ diurnal cycle, first two forward runs using the global model TM3 were performed. The differences in $CO_2$ simulations, extracted for a global set of measurement locations, were calculated by subtraction between

the daily- and hourly-based simulations. To derive the impact on retrieved fluxes, these differences in $CO_2$ simulations were then subsequently inverted using the standard CS configuration, except that all the prior fluxes and the initial $CO_2$ mole fraction in the model atmosphere were set to zero (while the prior uncertainty remained identical to that of the standard inversion). This

procedure is thus equivalent to using the difference of the results from two hypothetical inversions performed with and without diurnal cycle in the priors. That is, the linearity of the inversion operator is maintained in the inverted differences that represent the effect resulting from the hourly variability (deviations around daily means) required to be added to a daily mean, so as to reconstruct the diurnal cycle. The difference between the two inversions is thus determined by the transport operator and these hourly deviations, which lead to non-zero daily mean of simulated $CO_2$ mole fractions due to the rectifier effect. Therefore, a

correction of the magnitude of such a difference is required to be added to daily a posteriori flux estimated with missing hourly flux variations to reconcile the diurnal effect. These corrections are applied to CS flux estimates in this study as diagnostics. To find out the indirect impact on the regional flux estimates in the regional inversion CSR, a two-step inversion was done using the inverted differences in the "far-field contribution" from the global inversion.

## 3 Results

### 3.1 Differences in $CO_2$ mole fractions

We present results from the difference between the two forward runs performed with the transport model TM3 over a set of atmospheric stations distributed throughout the globe for the period of time 2010-2020 (the transport model was coupled with hourly and daily averaged biogenic fluxes that are obtained from FLUXCOM-X, see Sect. 2.4). The time-averaged differences in $CO_2$ mole fractions between daily and hourly simulations range from -2.49 to 6.97 ppm (1 ppm = 1 µmol mol$^{-1}$), Fig. 1.

Most of the sites (113) show positive differences with a mean of 0.67 ppm, while the remaining 34 sites resulted in negative differences with a mean of -0.41 ppm. These positive differences are found over sites that are more representative of the terrestrial land signal during day-time. On the other hand, the negative differences are dominated by some mountain sites where simulations are confined to a night-time window, resulting in a lagged land signal compared to day-time simulations. A portion of 27 sites of those 34 with negative difference are flask sites located either in remote islands or at shores. Therefore,

such sites are affected by large-scale ocean background with little terrestrial influences as well as by areas that contain mixed footprints from land and ocean, albeit with small influences. This implies that ignoring the diurnal cycle of $CO_2$ leads to an excess of $CO_2$ mole fraction over sites that are dominated by land footprints and to lower mole fractions over some mountain and ocean sites due to lagged land signal in comparison with the simulations done with the diurnal cycle included.

The differences due to the $CO_2$ diurnal cycle effect are also assessed separately for the different types of sites to distinguish their representativeness of land and ocean backgrounds, but also day- and night-time land signals. A large number of sites are located in lands and thus can have a good representativeness of the biosphere signal. Most of these sites are tall towers (33), continental sites (32), and surface sites (31), over which the largest positive mean differences in simulated $CO_2$ mole fractions

occur (0.70 ppm, 0.63 ppm, and 0.77 ppm, respectively, see Fig. 2, left panel). A number of 29 remote sites, mostly situated in islands, shows a very weak impact with a 0.02 ppm mean difference. A number of 13 mountain sites demonstrates a mean difference of -0.22 ppm indicating the dominant night-time land signal as simulations are restricted to a local night-time window over mountain locations. In addition, some sites that poorly represent the biosphere footprints such as baseline, ocean, and aircraft observing locations have quite small negative differences in $CO_2$ mole fraction simulations amounting to -0.10, -0.09 and -0.01 ppm, respectively. It should be noted that there are only 3 sites for each of these types, which makes it statistically difficult to draw conclusions, even though these small differences are expected over such type of locations where the biosphere signal is generally weak.

A large impact is observed during the growing season as the amplitude of the diurnal cycle reaches its maximum due to the strong uptake of $CO_2$ occurring under best conditions of light availability and soil water content. Figure 2 (right panel) shows monthly differences of $CO_2$ dry mole fractions averaged for eleven years (2010-2020) over 88 stations that dominate the diurnal impact in the Northern Hemisphere. All of these stations are towers, surface, or continental sites, which have a reliable representativeness of land footprints. The differences amid the analysed years range between 0.11 ppm and 1.37 ppm with a mean difference of 0.71 ppm. The median computed over these years is 0.70 ppm confirming a consistency with the mean. The large spreads of the monthly differences across years indicate a remarkable interannual variability of the diurnal impact. In June-July-August the mean difference amounts to 1.33 ppm, larger than the rest of the months, while the smallest differences were found in January-February-December with a mean of 0.38 ppm. The transition periods (March-April-May and September-October-November), close to the onset and termination of carbon uptake period, have moderate and relatively similar differences in $CO_2$ mixing ratios (0.85 and 1.0 ppm, respectively).

## 3.2 Differences in NEE estimates

To outline the impact of $CO_2$ diurnal cycle on the flux estimates, we next focus on analysing the differences in terrestrial NEE derived by the global CS inversion from the differences in $CO_2$ mixing ratios for the period 2010-2020. The diurnal cycle effect in the global annual budget of terrestrial biogenic fluxes estimated in the CS inversion results in a difference of 0.04 Pg C $yr^{-1}$ averaged over the analysed years with a standard deviation of 0.13 Pg C $yr^{-1}$. This difference in the global scale is rather small and equivalent to about 2 % of the mean annual terrestrial flux (-1.79 Pg C $yr^{-1}$), and only to around 1 % of the prior uncertainty assumed for the biogenic fluxes. These findings suggest that the diurnal cycle of $CO_2$ does not have a significant impact on the annual global flux budget as, due to the fact that global $CO_2$ flux estimates are well constrained by the growth rate of atmospheric $CO_2$ mole fractions, the diurnal cycle effect has to be compensated for between subregions and months. Even though there are noticeable seasonal variations in the impact of $CO_2$ diurnal cycle effect (Fig. 3), the negative differences during June-July-August are compensated by the positive differences during January-February-December and March-April-May when accounting for the impact on the annual scale. Differences during September-October-November remain around zero.

We next quantified the impact for latitudinal bands. Despite the negligible impact of the diurnal cycle at the global scale, the results indicate quite large differences in these bands (confined to NEE over lands, as ocean fluxes are not adjusted in the inversion set-up used here). Figure 4 illustrates that the largest differences are estimated between 90°S–15°S (0.39 Pg C yr$^{-1}$)

and between 15°S–15°N band (-0.46 Pg C yr$^{-1}$), on an order of magnitude similar to the original flux. In the bands 15°N–45°N and 45°N–90°N, the differences are smaller than the flux estimates but still non-negligible, with -0.17 and 0.27 Pg C yr$^{-1}$ change, respectively. As noted before, the overall difference over global land is quite small owing to the symmetry of differences in sign and magnitude along the latitude bands. The negative differences found in the bands that extend from equatorial to subtropical areas in the North (15°S–15°N and 15°N–45°N) are translated to stronger sinks in the corrected flux

budgets. On the other hand, the bands containing temperate and boreal zones (90°S–15°S and 45°N–90°N) show positive differences, which imply additional sources in the corrected flux budgets.

Note that the size and even the sign of the effects quantified here may depend on the size and distribution of diurnal variations in the flux data set employed in the forward runs, as well as on the inversion set-up and choice of stations. Therefore, they are meant as diagnostics for the diurnal effect of $CO_2$. Even if the corrections of the flux budgets applied in this paper are denoted

in Figures as "CorrectedBudget", they are tentative changes based on the specific diurnal cycles from FLUXCOM-X and the specific inversion set-up and station set used here.

The excess of simulated $CO_2$ mole fractions resulting from using daily NEE makes the inversion adjust to stronger sinks as compensatory fluxes, while the underestimation of simulated $CO_2$ mole fractions (seen at only a few sites) is compensated by increasing sources. Basically, the correction to be added to the inversion using daily mean priors is an inversion of the

difference of daily - hourly mean prior fluxes. Therefore, the positive differences shown in Fig. 4 lead to a weaker sink (or additional $CO_2$ sources) in the posterior fluxes and vice versa for the negative differences.

To investigate the aforementioned compensation effect in a set of subregions across the globe, we analysed the differences calculated with CS over the set of regions used in the TransCom experiment (Gurney et al., 2003). Figure 5 indicates large

changes in the annual flux budgets over most of the regions. As expected, the results exhibit positive and negative differences over land, leading to the compensation in the annual mean difference and flux estimates over the globe. In this context, negative differences imply either an underestimation of $CO_2$ uptake or overestimation of $CO_2$ release by inversions when neglecting the diurnal cycle in the flux. Negative differences are found for Northern and Southern Africa, Eurasian temperate forests, North American boreal forests, and tropical Asia. On the other hand, positive differences are found over North American temperate

forests, South American tropics and temperate, Eurasia boreal, Australia, and Europe. This implies, additional $CO_2$ sources are suggested in the flux estimates for the regions exhibiting positive differences, while flux estimates of the regions with positive differences should be corrected by allocating more sinks. This, to some extent, modifies the dipoles (source-sink compensation) persistent in the inversion estimates reported in literature, particularly northern extratropics versus south and tropical lands (Friedlingstein et al., 2023; Kondo et al., 2020; Peylin et al., 2002). In this case, the inversions tend to allocate

stronger uptake in northern extratropical lands relying on the gradient of observations distributed across the latitude bands at

the expense of $CO_2$ source allocations placed in tropics and southern lands, where observational coverage is actually poor, to maintain the global atmospheric $CO_2$ growth. To the limit of the results shown in Fig. 5, the large increase of sinks due to the diurnal effect suggested in Africa corresponds to reallocating large sources in north and south American temperate forests, south American tropics, and Australia. Although such additional sources are distributed over Southern and Northern

Hemispheres, the stronger uptake in Africa moderates the dipole effect in the results of CS. Additionally, the opposing flux corrections calculated between Eurasian boreal and temperate forests cancel out the effect on dipoles if the flux estimates over both regions are aggregated together.

Moreover, the largest negative difference (-0.79 Pg C yr$^{-1}$) among the TransCom regions is found in Northern Africa, which would turn the flux budget from a net source of 0.50 Pg C yr$^{-1}$ to a net sink of -0.29 Pg C yr$^{-1}$. Over Temperate North America,

a large positive difference of 0.38 Pg C yr$^{-1}$ is found, pushing the flux budget from a net source (about 0.15 Pg C yr$^{-1}$) to a larger net source (0.53 Pg C yr$^{-1}$). Given the positive and negative differences seen in "Diff" Fig. 5, the mean annual difference for the global scale stays roughly around zero globally (see above). Generally, the posterior fluxes estimated by CS without the diurnal cycle of $CO_2$ include biases that require corrections by additional sinks over regions that show negative differences and of additional sources over regions characterised with positive differences. These findings demonstrate that the systematic

biases in the annual flux budgets get larger when disaggregating the total land to the continental scale.

### 3.3 Analysis of the diurnal cycle effect of $CO_2$ over the European continent

This section will address results from the global CS and from CSR for the domain used for the regional inversion covering most of Europe (15°W-35°E, 33°S-73°N). It should be noted that this CSR-domain is different than the one used for Europe

in the TransCom set of regions in CS in Fig 5, and therefore NEE differences and flux estimates are expected to be slightly different over both domains. The differences for this domain estimated with CS due to missing the $CO_2$ diurnal cycle in CS amounts to 0.12 Pg C yr$^{-1}$, averaged over the period 2010-2020. It represents about 25% of the prior NEE uncertainty assumed for annual fluxes for this area, but also exceeds the posterior uncertainty by a factor of 2. Furthermore, the mean flux budget computed for the respective years was –0.37 Pg C yr$^{-1}$. Thus, this diurnal effect leads to a bias of around 32% in the annual

flux estimates in Europe. There are also slight monthly variations of the impact seen among years (Fig. 6, "CS").

To assess the diurnal effect on the regional inversion for Europe, CSR was used to estimate the differences in NEE due to the diurnal cycle effect passed on from the global inversion via the far-field contributions calculated by CS. The mean differences over all the years appear to be consistent in both the magnitude and temporal patterns between CSR and CS estimates, albeit larger in CSR. The mean difference calculated with CSR results in 0.15 Pg C yr$^{-1}$, representing 32% of the prior uncertainty

assumed in CSR. When relating the differences to posterior uncertainty calculated with CSR over Europe, the impact even exceeds the magnitude of posterior uncertainty by a factor of around 2.5. Additionally, the $CO_2$ diurnal effect leads to a bias of around 48% in $CO_2$ estimates (-0.31 Pg C yr$^{-1}$) that were calculated as the mean annual flux budget for the respective years by CSR.

Although there are notable variations in monthly differences of CSR over years shown in the error bars reflecting the spread over years (see Fig. 6, "CSR"), they agree in the magnitude and in month-to-month variations with those of CS. The seasonality in differences suggests a compensation effect when integrating NEE estimates to the annual scale. Additionally, the findings demonstrate a smaller year-to-year variability of the $CO_2$ diurnal effect during winter compared to the variability during spring and summer. This leads to dominant contributions of spring and summer fluxes to the interannual variability of NEE estimates resulting from the $CO_2$ diurnal cycle.

### 3.4 How much does the $CO_2$ diurnal cycle affect inter-annual variability?

Figure 7 illustrates the annual flux differences due to the diurnal cycle effect of $CO_2$, and the corresponding flux estimates. The findings indicate how much the lack of the $CO_2$ flux diurnal cycle affects annual $CO_2$ flux budgets for the individual years, globally and regionally over the CSR-Europe domain. Including the diurnal cycle of $CO_2$ generally shifted estimates towards sources over all the years, albeit in different magnitudes. The analysis here confirms that the interannual variability (IAV) of flux estimates is less sensitive to the diurnal variations at the global scale ("Global", Fig. 7). Although the mean annual flux differences between daily and hourly NEE-based inversions stay around zero over the analysed years, they can be slightly larger in individual years (standard deviation of 0.13 Pg C $yr^{-1}$). Furthermore, the similarity of IAV for the annual flux estimates and the corrected estimates (0.82 Pg C $yr^{-1}$ and 0.88 Pg C $yr^{-1}$, respectively) indicates the negligible influence of $CO_2$ diurnal variations on the IAV of global estimates. By contrast, the results demonstrate a larger impact on the IAV of the regional flux estimates over Europe. The NEE differences calculated by CS resulted in an IAV amplitude of -0.06 Pg C $yr^{-1}$, about half the IAV amplitude of the estimated fluxes (-0.14 Pg C $yr^{-1}$).

In terms of the indirect effect of diurnal variations passed to the regional inversion through far-field influences, CSR suggests a slightly weaker uptake (-0.16 Pg C $yr^{-1}$) than CS does (-0.25 Pg C $yr^{-1}$) after taking into consideration the diurnal cycle effect. The IAV of the differences over Europe calculated with CSR amounts to 0.09 Pg C $yr^{-1}$, suggesting a significant impact when comparing it with the IAV of the annual flux estimates 0.20 Pg C $yr^{-1}$. Noteworthy, CS and CSR differ in their atmospheric transport models, and the spatial correlation of prior uncertainty are chosen differently. Consequently, such different setups are likely resulting in the discrepancies between CS and CSR in evaluating the impact of the $CO_2$ diurnal cycle.

To show the spatial patterns of the differences in the domain of Europe, posterior NEE estimated in a regional (CSR) inversion is analysed together with the differences due to the diurnal variations in CS and the corrected estimates. Figure 8 depicts the annual mean of NEE without and with the diurnal cycle of $CO_2$ taken into account, and the differences, averaged over eleven years. Positive corrections to NEE fluxes occur generally along the Mediterranean coast (southern and western Europe), as well as north-western Europe (southern UK, Benelux, northern Germany). On the other hand, over Scandinavia, northern UK, and in central Europe smaller negative corrections (meaning higher $CO_2$ uptake in the respective regions) are persistent. These findings refer to a notable impact of $CO_2$ diurnal cycle on NEE estimated over smaller local domains, particularly over regions where $CO_2$ exchange between the atmosphere and terrestrial ecosystems is more active. It should be noted that because of the

partial compensation between subregions with positive and negative differences, the annual mean difference in NEE fluxes will also become smaller as larger areas are aggregated, underlining the increasing importance of the $CO_2$ diurnal effect as finer spatiotemporal scales are analysed.

## 4 Discussion

The seasonal features of the covariation between atmospheric transport convection and terrestrial $CO_2$ fluxes discussed in (Denning et al., 1995; 1996b; 1996a) hold true for the diurnal variability of both the atmospheric dynamics and biota. Ignoring the diurnal cycle of $CO_2$ in the biosphere fluxes used as priors in inversions results in significant biases in $CO_2$ mole fraction simulations calculated with atmospheric transport models. In this study, we quantified the effect on the $CO_2$ mole fraction and estimated NEE arising from the diurnal cycle of $CO_2$ at global and regional scales. We find that $CO_2$ mole fractions simulated during day-time at sites that are dominated by land footprints with the inclusion of the biogenic diurnal variability tend to be lower than those simulated with daily averaged biogenic fluxes; the opposite effect is found for the simulations calculated during night-time over mountain sites, as well as over sites that are more representative for ocean backgrounds where the biosphere signal is generally weak and lagged. Also, the impact found over locations with dominant land backgrounds that also have a larger number of sites was much larger in magnitude than that found over locations with ocean backgrounds. This typically points to the prevalence of the diurnal effect over lands, particularly in the biogenic terrestrial ecosystems in the northern Hemisphere.

The differences in simulated $CO_2$ mixing ratios inevitably lead to differences in NEE estimates derived from atmospheric inversions. Regarding the global total flux, the impact of ignoring the diurnal cycle is compensated for by the seasonal and spatial variations, as the inversion constrains reasonably well the long term mean $CO_2$ flux irrespective of when and where the flux adjustments are allocated. This is evident by the small annual difference of 2 % of the global land flux estimate (-1.37 Pg C $yr^{-1}$), which lies within the range of total atmosphere-to-land sinks (1.10 and 1.70 Pg C $yr^{-1}$) estimated by inversions as reported to the Global Carbon Budget (GCB) in Friedlingstein et al. (2022), to which CS contributes. Consequently, neither IAV nor the magnitude of global flux estimates has significantly changed after applying corrections of the $CO_2$ diurnal effect over the analysed period of time, indicating a negligible impact in the integrated global flux budget.

However, the diurnal cycle of $CO_2$ does influence NEE estimates calculated at regional domains. For example, results analysed over the land region within the CSR domain covering Europe yield an NEE difference of 48% and 32% of the annual estimates calculated by CSR-Europe and the global CS, respectively. The annual mean differences represent a shift of flux estimates towards higher $CO_2$ sources in both inversions. Even though the diurnal cycle is taken into account in CSR within the regional domain by using biogenic prior fluxes obtained from biosphere flux models at hourly time intervals, the differences from the global CS are passed to CSR estimates as an indirect effect through the lateral boundary conditions. By comparing two products of lateral boundary conditions obtained from CS and TM5-4DVar global inversions, Munassar et al. (2023) also found a non-negligible sensitivity of regional inversions to systematic biases in lateral boundary conditions. They found an impact of 0.40

Pg C yr$^{-1}$ resulting from the lateral boundary conditions over the CSR-domain of Europe for 2018. As the diurnal cycle of $CO_2$ was missing in the setup of CS used in that study, part of this impact can potentially be attributed to the diurnal cycle effect. This suggests that the systematic biases in regional inversions due to lateral boundary conditions would be attenuated to approximately 0.25 Pg C yr$^{-1}$ if taking into account a bias of 0.15 Pg C yr$^{-1}$ arising from the diurnal cycle effect found in CS via this study.

The problem of the seasonal rectifier considered by Denning et al (1995) could be solved by explicitly estimating seasonal variations from the atmospheric data. Unlike this, unfortunately, the diurnal cycle cannot be deduced from the atmospheric data, primarily owing to the limitation of the assimilated data to either day-time or night-time. Even though a weak variability may emerge due to the diurnal variability of atmospheric vertical transport, it cannot reproduce a realistic rectification observed in atmospheric $CO_2$ concentrations (Yi et al., 2000) without coupling the transport variability with a-priori information of terrestrial flux variability. In an experiment done at a site level with and without diurnal biosphere variations, (Denning et al., 1996a) found that diurnal $CO_2$ concentrations simulated with diurnal flux variations were more realistic in phase when compared to observations, while those simulated with mean prescribed biogenic fluxes follow the phase of the PBL depth. This indicates the importance of accounting for the diurnal variations when retrieving atmospheric $CO_2$ dry mole fractions. Further assessment of the uncertainty of such variations produced from biosphere models will benefit the characterization of estimated $CO_2$ flux errors among the inversions contributing to the Global Carbon Project (GCP), but also extend the understanding of biogenic terrestrial variability. Notwithstanding, the FLUXCOM-X model, used in our experiment, is thought to have a good ability to resolve the diurnal cycle of $CO_2$ because the model is trained against observations, meteorology, and remote sensing data using different prediction approaches to reproduce GPP at half-hourly time scales (Bodesheim et al., 2018; Nelson et al., 2024). This is supported by a validation analysis by Nelson et al. (2024) using the FLUXCOM-X dataset. The outcome of this analysis suggests that FLUXCOM-X provides good pieces of information regarding the diurnal variations, in spite of some shortcomings concerning the seasonal and interannual variabilities. Such flux products can thus be beneficial for global inversions to provide a better constraint for $CO_2$ diurnal cycle, provided that seasonal and interannual variations are already constrained by $CO_2$ observations. Nevertheless, the eddy covariance measurements needed to estimate and validate diurnal cycles in upscaling products like FLUXCOM-X are known to represent only a certain part of the biosphere, thus the representativity of the upscaled fluxes on continental scales is hard to assess.

In a further assessment, the differences in NEE due to the $CO_2$ diurnal effect were analysed (over three latitude land bands representing southern hemisphere, tropics, and northern hemisphere) using biosphere fluxes from the Carnegie-Ames Stanford Approach (CASA) (Potter et al., 1993) instead of FLUXCOM-X (Fig. 9). The biogenic flux products of CASA are used by – and provided through – the Carbon Tracker inversion CT2022 (Jacobson et al., 2023), where the temporal downscaling of monthly Net Primary Productivity (NPP) and ecosystem respiration is performed using surface solar radiation and temperature from ERA5 reanalysis following the methodology by Olsen and Randerson (2004) to include the diurnal cycle of $CO_2$. The differences in NEE calculated with CASA-based fluxes were estimated by CarboScope using the same way as the differences estimated with FLUXCOM-X, as described in the methods section. Quite small differences were found between daily and

hourly fluxes calculated with CASA compared to those calculated with FLUXCOM-X over the three latitude bands. This implies that the $CO_2$ diurnal cycle retrieved using Olson-Randerson method is much weaker than that calculated with FLUXCOM-X. Noteworthy, the Olson-Randerson method is used by several global inversions to produce diurnal variations for the biogenic fluxes if not yet represented by the biosphere flux models themselves used as priors. Therefore, such discrepancies in resolving the $CO_2$ diurnal cycle among biosphere models suggest that more work is needed regarding the best methodology to be applied to generate the diurnal variations.

In the tropics (30°S-30°N) the difference was found to be -0.65 Pg C $yr^{-1}$ over land, proposing a much stronger sink (-1.48 Pg C $yr^{-1}$) than CS estimated (-0.83 Pg C $yr^{-1}$). Friedlingstein et al. (2022) found that atmospheric inversions suggested the land tropics to be close to neutral over the past decade ranging between -0.90 and 0.70 Pg C $yr^{-1}$ with high uncertainty. After considering the impact of diurnal cycle, CS estimates for the tropical lands will tend to result in even stronger $CO_2$ sinks. On the contrary, latitude bands containing temperate forest zones show weaker $CO_2$ uptake when the correction due to the diurnal cycle is taken into account. That is, the land sink over the northern extra-tropics (30°N-90°N) amounts to -0.53 Pg C $yr^{-1}$ after correcting for the diurnal cycle effect compared to -1.12 Pg C $yr^{-1}$ estimated with CS for 2010-2020. Hence, the underestimation of $CO_2$ uptake in the tropics and the overestimation of $CO_2$ uptake in temperate zones are the trade-off that maintains $CO_2$ mass-balance in the global carbon budgets derived by CS.

Given the non-negligible impact of the diurnal cycle of $CO_2$ over the latitude bands, the impact is quantified in the context of annual flux budgets at continental and regional scales using CS results over the globe. NEE differences were analysed over a set of regions, encompassing the whole land area around the world, as used in the TransCom experiment (Gurney et al., 2003), and are shown in Fig. 5. The differences over most of the land regions exhibit large biases in the posterior annual fluxes over the analysed regions. This has been outlined by applying flux corrections that suggest additional sources in the posterior annual fluxes over some regions such as the temperate north America region, while stronger sinks are suggested over other regions as is the case of northern Africa. The differences of the two largest cases in northern Africa and in temperate north America (-0.79 and 0.38 Pg C $yr^{-1}$, respectively) exceed the corresponding uncertainties (0.54 and 0.22 Pg C $yr^{-1}$, respectively) calculated as the spread among different model estimates reported in Gurney et al. (2004) within the TransCom Experiment. The global mean difference is, however, small and consistent with the global total uncertainty. These findings indicate that misallocations of flux adjustments in different regions occur as a result of lacking diurnal information in the atmospheric inversions. Noteworthy, some regions that show large differences such as Africa are repeatedly characterised with a larger uncertainty in inversion estimates by previous studies (e.g., ). These studies attribute the compensatory fluxes over such regions to the lack of observational constraints and thus may be allocated with the surplus of flux budgets remaining from the well-constrained regions. Moreover, estimates of terrestrial $CO_2$ fluxes by inversions reported in other studies generally suffer from larger uncertainties over north and south America and Africa. This is in principle attributed partially to the lack of observational constraint, as reported by Byrne et al. (2023), in which several global inversions were utilised with different setups of atmospheric dataset sources obtained from in situ and OCO-2 satellite measurements in the Model Intercomparison Project (MIP) to estimate Net Carbon Exchange (NCE). However, the inversions in that study included prior biosphere fluxes from

various models that have different prescription of estimating the diurnal variations such as CASA, SiB-4, ORCHIDEE and CARDAMOM. The posterior estimates are likely to be driven by prior flux models, especially over tropical regions as a consequence of not having enough observations. This is reflected in the increase of spatial uncertainty estimated by Byrne et

al. (2023) across tropical and subtropical regions, where substantial differences in NEE estimates are found in our study due to the diurnal $CO_2$ effect. Therefore, the substantial discrepancies in posterior $CO_2$ fluxes calculated using global inversions over regional scales, e.g., north and south America and Africa as reported by Friedlingstein et al. (2023), are not only caused by transport and lack of observations but also involve a contribution from inaccuracies in representing $CO_2$ diurnal variations estimated by bottom-up biosphere models used as priors in Bayesian inversions.

NEE differences due to the diurnal cycle effect show year-to-year variations that affect IAV of the flux estimates. This is evident by the sensitivity of NEE IAV to the diurnal cycle of $CO_2$ assessed over the TransCom land regions (Table 2). Regions with a large mean difference in NEE also tend to have a larger IAV of the diurnal cycle impact, such as northern Africa, north American temperate, and south American tropics, but also the rest of the land regions suggest a large IAV in NEE differences. This finding implies that both the magnitude of NEE estimates and IAV are affected by the neglect of $CO_2$ diurnal cycle at

regional scales. Noteworthy is that this analysis is performed with a global in situ site network using 147 sites, representing about the total number of stations available to provide measurements to normal CS inversions. Hence, the IAV can be subject to changes if the analysis was made with different stations set and with different temporal coverage of dataset over the analysed period 2010-2020. Even though evaluating the uncertainty of diurnal variations remains an important target for future work, the inclusion of such variations in biogenic prior fluxes used in inversions should generally decrease biases in $CO_2$ mole

fraction simulations and thus reduce biases in estimated $CO_2$ fluxes by inversions.

Based on the results, the stronger uptake dominant in the northern hemisphere attributed to the $CO_2$ fertilisation effect and deposition of nitrogen (Ciais et al., 2019; Sarmiento et al., 2010) cannot be predicted accurately by inversions that do not include diurnal cycle of $CO_2$ in their prior fluxes. The greatest impact of the $CO_2$ diurnal effect obviously appears over regions with strong biological activities as seen from a diagnostic map of $CO_2$ mole fractions simulated at the model surface level with

the inverted response of the diurnal effect across the world (Fig. 10). Averaging out the diurnal variations in the biosphere flux model FLUXCOM-X has led to overestimation of $CO_2$ mole fractions, largely over the temperate areas where a large amount of carbon is stored in the temperate forests in North America, Europe, Eastern Asia in the Northern Hemisphere as well as over the rainforests of South America and Australia in the Southern Hemisphere (Erb et al., 2018).

By contrast, CS tends to underestimate $CO_2$ mole fractions as the $CO_2$ diurnal cycle of FLUXCOM-X is flattened in the tropical areas, in particular central Africa and south eastern Asia. This pattern is also, to a lesser extent, seen over the boreal forests in the further North. These results outline the discrepancies in the annual flux budgets computed over the TransCom regions and over the latitude bands. It should be noted that these corrections are used in this study as diagnostics as the diurnal cycle is subject to uncertainty depending on the accuracy of the biosphere model to reproduce diurnal variations. Additionally, the

spatial patterns of mole fractions over Europe in Fig. 10 correspond quite well to the spatial patterns of posterior flux differences estimated by CSR (Fig. 8).

## 5 Conclusions

Ignoring the diurnal cycle in surface-to-atmosphere $CO_2$ fluxes leads to a systematic bias in $CO_2$ mole fraction simulations sampled at daytime, because the daily mean flux systematically misses the $CO_2$ uptake during the daytime hours. In an
atmospheric inversion using daytime-selected $CO_2$ measurements at most continental sites and not resolving diurnal cycles in the flux (as is the case for the 60-year global CarboScope $CO_2$ inversion that operates on a daily flux time step due to limitation in computer memory), this leads to systematic biases in the estimates of the annual sources and sinks of atmospheric $CO_2$. In case of the absence of diurnal variations in the prior inputs, correcting the flux estimates is essential to reduce these systematic biases. Such a correction can be applied either to the mole fractions before performing the inversions or to the optimised fluxes
after performing the inversions. In this paper, we diagnosed how large such a flux correction would need to be in annual carbon flux budgets derived from CS and CSR for 2010-2020 at global and regional scales. NEE containing the diurnal cycle of $CO_2$ flux was obtained from the data-driven biosphere flux estimate by FLUXCOM-X (Nelson et al., 2024) at hourly time scales over the globe. These fluxes were transported by the TM3 model to simulate $CO_2$ mole fractions. Removing the diurnal cycle of NEE by using daily fluxes leads to an increase of simulated $CO_2$ mole fractions in comparison with simulations done with
hourly NEE, because of the diurnal effect described above. The difference between both simulations was inverted to obtain the impact of this effect on the inversion results. Regarding the net fluxes of $CO_2$ estimated by the global CarboScope (CS) inversion, the difference in the flux budget due to the diurnal cycle was negligible at the global scale, which is expected as the global annual trend of $CO_2$ is well-constrained by the observations. However, based on FLUXCOM-X diurnal cycles, the impact at the regional and local scales amounts to up to about 51% relative to the sum of differences and flux estimates,
depending on region. The analysis of NEE differences across latitudinal bands and for the set of regions used in the TransCom experiment points to larger differences exceeding the magnitude of the estimated annual flux budgets in some regions such as North American temperate and Northern Africa. The overall NEE differences in the TransCom land regions suggest a weaker sink in temperate zones (particularly in the North) and a stronger sink in parts of the tropics, than estimated when ignoring the diurnal cycle. This illustrates a compensation effect between the regions to retain global mass balance as discussed above. In
addition to the mean flux, also the IAV of flux estimates is affected by the diurnal variations. This indicates that it may be unrealistic to use climatological diurnal variations to correct the atmospheric $CO_2$ inversions, though further investigations are required to look into the origin of IAV in the NEE differences (e.g., to determine how much biases come from the amplitude of the diurnal cycle of $CO_2$ in the biosphere model and from BPL in the transport model).

When replacing FLUXCOM-X diurnal cycles by diurnal cycles according to the Olsen and Randerson (2004) method, our
analysis results in a substantially smaller impact on the inversion results. The Olsen and Randerson (2004) method is used in

several global inversions taking part in the Global Carbon Budget of Friedlingstein et al. (2023). Hence, a more comprehensive assessment on the uncertainty of the diurnal cycle effect among atmospheric inversions is planned in a follow-up study.

NEE estimates derived from a regional inversion, even if using a diurnal cycle in its regional prior, are still prone to an indirect impact of the diurnal cycle effect through the lateral boundary conditions if coming from a global inversion that is missing diurnal variations. In our case of CarboScope-Regional (CSR) for Europe using boundary conditions from the global CS, such an impact amounts to approximately 50% of the mean annual flux estimates calculated over Europe.

We conclude that incorporating diurnal variations into the prior fluxes, directly or via a suitable correction, is important in global and regional atmospheric $CO_2$ inversions. Further work is needed to assess the uncertainty of the effect, and to develop a suitable implementation of the diurnal cycle effect into the global CarboScope $CO_2$ inversion.

### Code and data availability

The simulations of $CO_2$ mole fractions, NEE differences derived from the inversions, and codes used for the analysis can be made available upon request to the corresponding author.

### Author contributions

SM, CG, and CR designed the study. SM performed the simulations and analysis and drafted the paper with contributions from CR. CG, KUT, MG, FTK, and SB provided support. All the authors revised the paper and edited the text.

### Competing interests

At least one of the (co-)authors is a member of the editorial board of Atmospheric Chemistry and Physics and the authors also have no other competing interests to declare.

### Acknowledgments

The authors thank Sönke Zaehle for his valuable comments on the manuscript in the internal review. We also acknowledge the use of hourly NEE data calculated with FLUXCOM-X provided by Jake Nelson and Sophia Walter; CarbonTracker CT2022 and CT-NRT.v2023-3 results provided by NOAA GML, Boulder, Colorado, USA from the website at http://carbontracker.noaa.gov. This work used resources of the Deutsches Klimarechenzentrum (DKRZ) granted by its Scientific Steering Committee (WLA) under project ID bm1400.

### Financial support

This work is funded by the German Federal Ministry of Education and Research (BMBF) project "Integrated Greenhouse Gas Monitoring System for Germany – Modellierung (ITMS M)" under grant number 01 LK2102A.

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

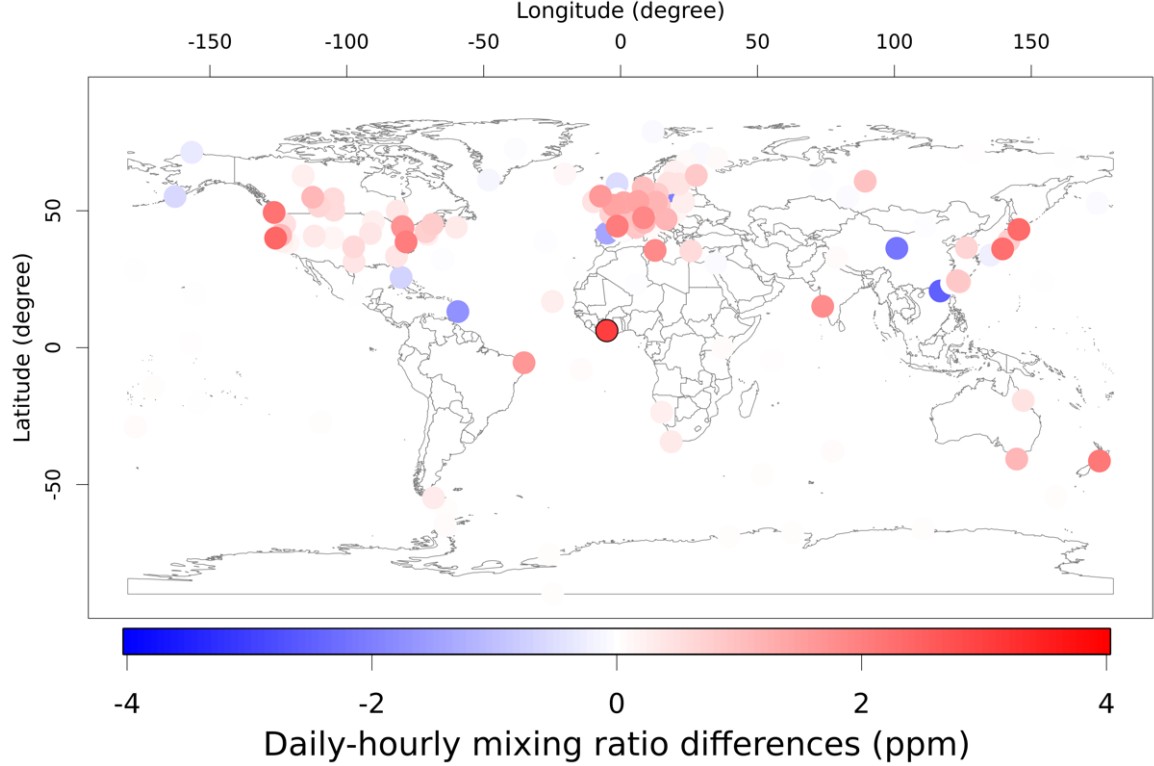

**Figure 1: CO$_2$ mole fraction differences between daily and hourly NEE-based simulations averaged for 2010-2020. Note, the difference at the site with a black circle is 6.97 ppm, excluded on the legend range for the visibility of other sites with smaller values.**

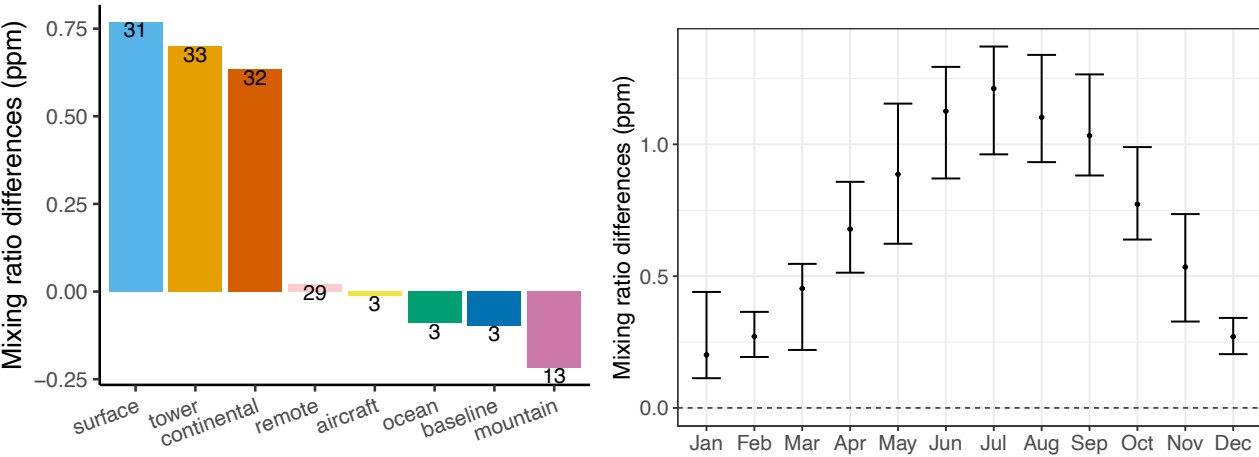

**Figure 2: Differences of CO$_2$ dry mole fractions between simulations calculated with daily- and hourly-based NEE over specific site classifications and over months. Left panel shows the mean differences averaged over site-specific classifications (on x-axis) for the analysed years (2010-2020), and the numbers mentioned in bars are the number of sites per each classification; right panel denotes monthly differences computed for the northern Hemisphere and averaged over 88 sites representing towers, surface, and continental stations that dominate the impact over northern hemispheric lands (note: error bars refer to the range of differences over the target years (2010-2020).**

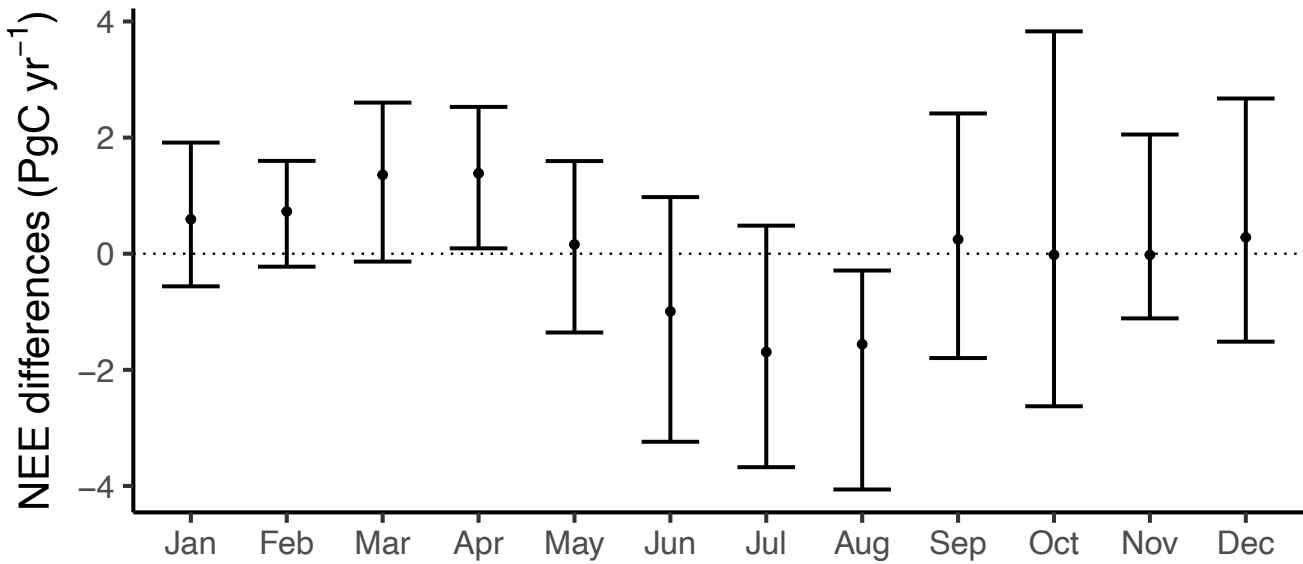

**Figure 3: Monthly mean differences in global NEE estimates resulting from $CO_2$ diurnal cycle, averaged over 2010-2020. Error bars refer to min and max differences among all the analysed years.**

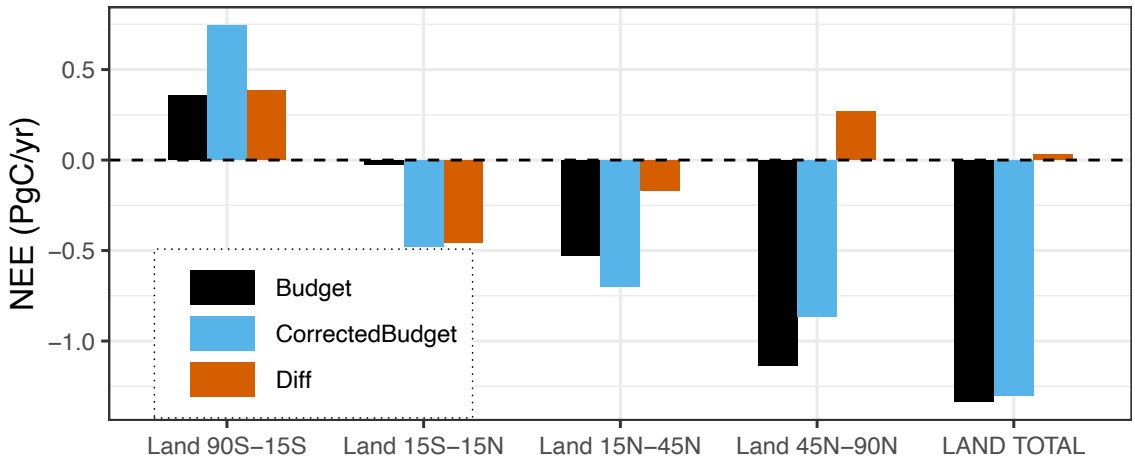

**Figure 4: Integrated annual differences of NEE along latitude bands; NEE estimates calculated with CS are also shown before and after the corrections due to the diurnal cycle.**

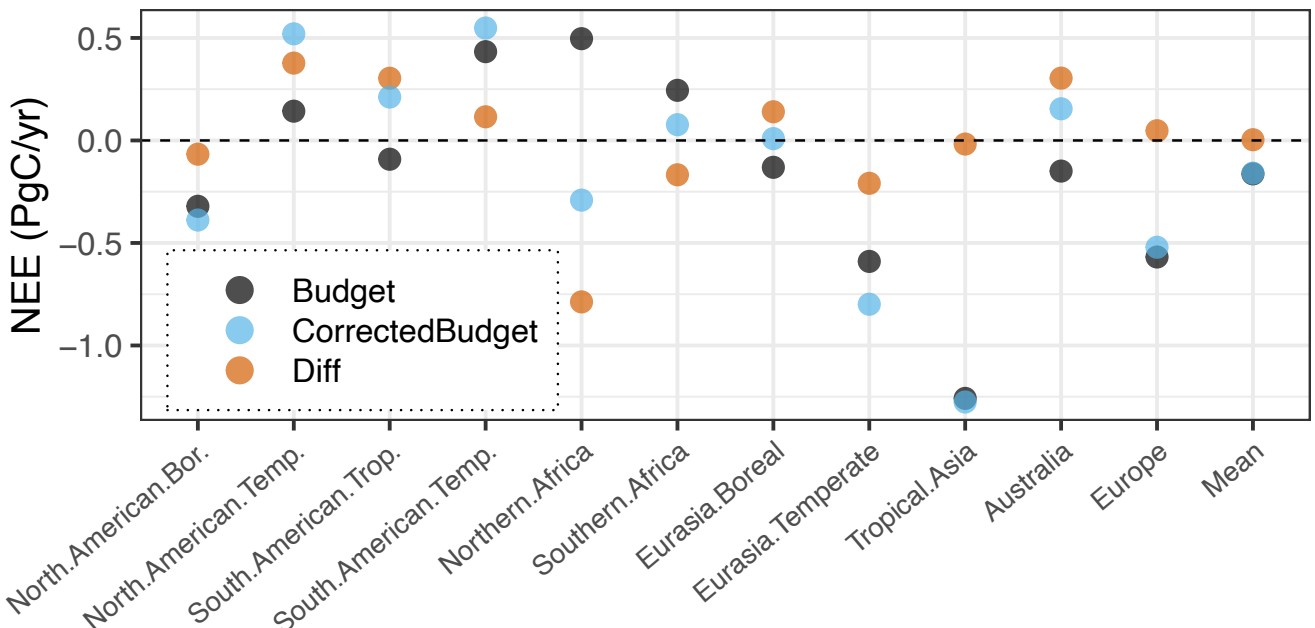

**Figure 5: Annual NEE estimates and their respective corrected estimates based on differences due to diurnal cycle effect integrated over TransCom regions, averaged over the time period 2010-2020.**

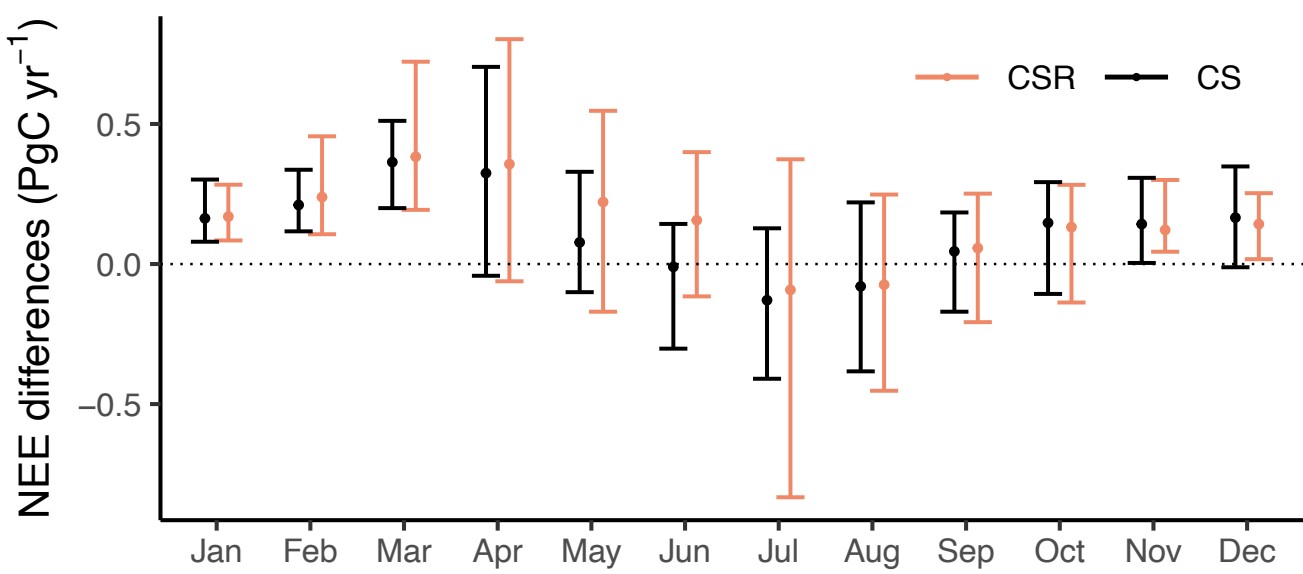

**Figure 6: Monthly mean NEE differences resulting from $CO_2$ diurnal cycle with CSR (red) and CS (black) estimated over Europe using the differences in mixing ratios for the period of time 2010-2020. Error bars mark the range of monthly differences over the analysed years.**

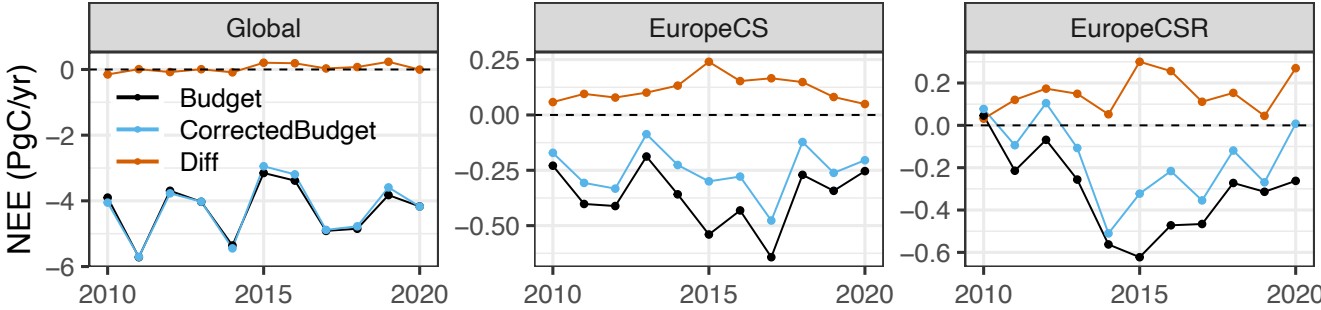

**Figure 7: Annual flux budgets (before and after corrections due to CO₂ diurnal cycle) estimated using CS over the globe and Europe using CS and over Europe using CSR for 2010-2020.**

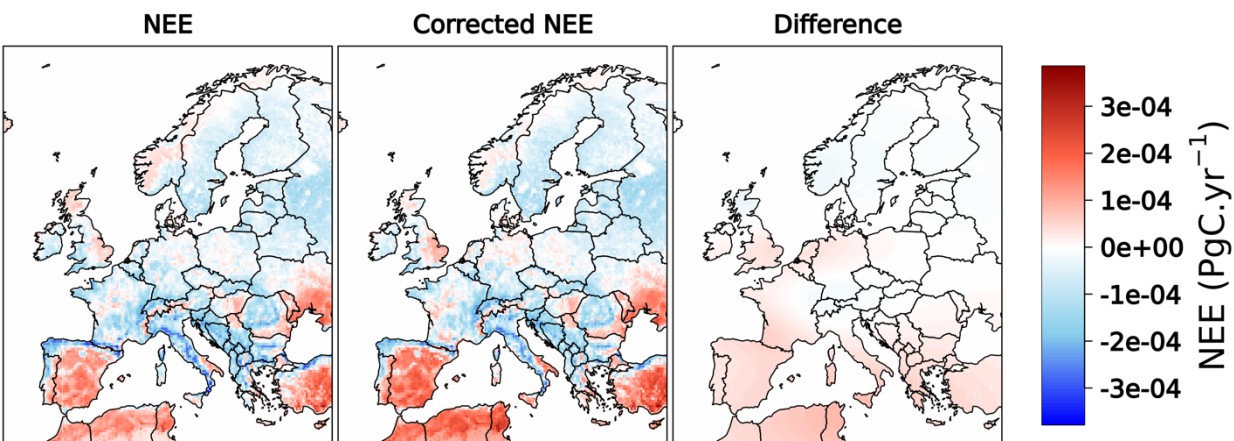

**Figure 8: Spatial distributions of NEE, corrected NEE, and differences (from left to right, respectively) calculated as the mean of 2010-2020 over Europe**

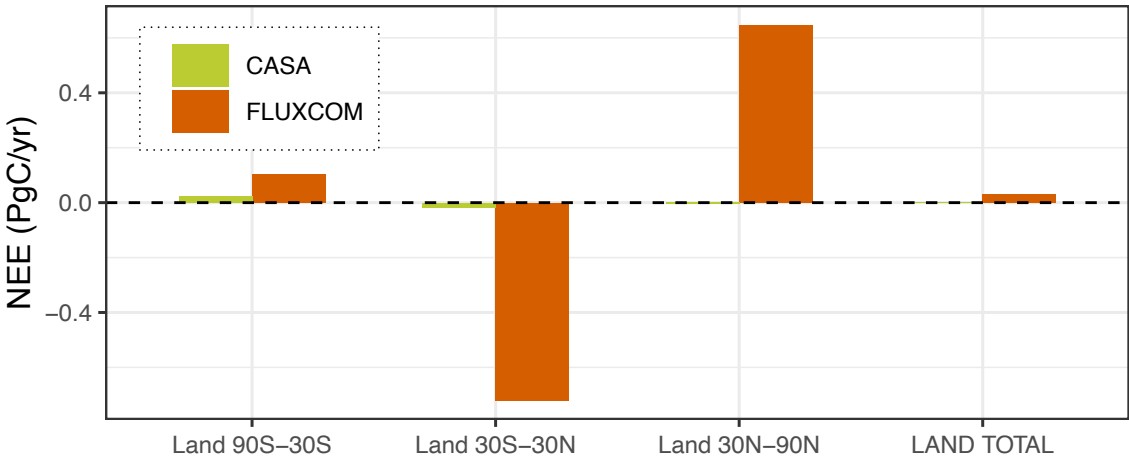

**Figure 9: Differences in NEE resulting from the diurnal cycle effect of $CO_2$ estimated by CarboScope using two biosphere models (CASA-based biogenic fluxes in green bars and FLUXCOM-X in red bars), averaged over 2010-2020 across a set of latitude land bands indicated on x-axis.**

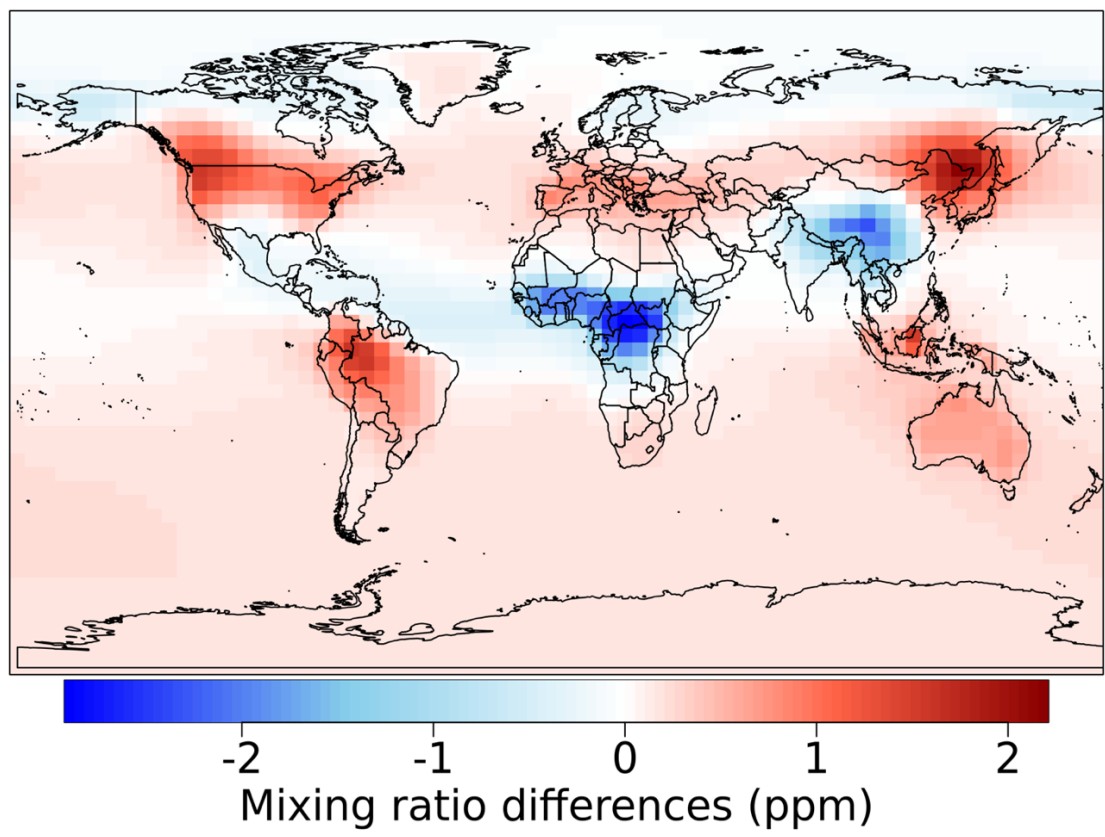

**Figure 10: 2-D fields of CO$_2$ mole fractions simulated using the inverted response of the diurnal cycle (i.e., inverted differences of mole fractions calculated with daily- and hourly- based NEE) for local daytime at the model surface level for the period 2010-2020 over the globe with TM3**

**Table 1: CS and CSR inversion setups**

| Inv. | Domain | Transport model | Diurnal CO$_2$ flux | Unc. shape | Unc. structure | Spatial resolution of state space |
|------|--------|-----------------|---------------------|------------|----------------|-----------------------------------|
| CS | global | TM3 (5°x4°) | no | Flat | exponential | 2.5°x2° |
| CSR | Europe | STILT (0.25° x 0.25°) * | yes | Flat | hyperbolic | 0.5°x0.5° |

*\* - resolution of the driving meteorological fields; STILT is a lagrangian particle model simulating subgrid scale vertical mixing at effectively higher spatial scales.*

**Table 2: Sensitivity of IAV to the impact of $CO_2$ diurnal cycle. "Diff. IAV" corresponds to the IAV of the impact of diurnal cycle on retrieved fluxes, "Flux IAV" corresponds to the IAV of the estimated fluxes themselves, and "Flux IAV corr." corresponds to IAV of estimated fluxes after corrections.**

| Land region (TransCom) | Diff. IAV (Pg C yr$^{-1}$) | Flux IAV (Pg C yr$^{-1}$) | Flux IAV corr. (Pg C yr$^{-1}$) |
|---|---|---|---|
| North.American.Bor. | 0.04 | 0.12 | 0.14 |
| North.American.Temp. | 0.13 | 0.10 | 0.18 |
| South.American.Trop. | 0.29 | 0.20 | 0.39 |
| South.American.Temp. | 0.18 | 0.25 | 0.35 |
| Northern.Africa | 0.59 | 0.23 | 0.73 |
| Southern.Africa | 0.17 | 0.17 | 0.26 |
| Eurasia.Boreal | 0.07 | 0.32 | 0.35 |
| Eurasia.Temperate | 0.23 | 0.23 | 0.28 |
| Tropical.Asia | 0.21 | 0.18 | 0.21 |
| Australia | 0.12 | 0.11 | 0.19 |
| Europe | 0.08 | 0.17 | 0.24 |

895