# Peer review of "To what extent does CO2 diurnal cycle impact flux estimates derived from global and regional inversions?"

_EGUsphere, 2024_

## Author Comment (AC1)

*Point-by-point response to Referees #1 and #2*

*First of all, we very much thank the anonymous Referees #1 and #2 for the interesting comments and suggestions that indeed help improve the manuscript. In the following sections, we address the comments, suggestions, and concerns of the Referees point-by-point throughout this document. Additionally, the corresponding modifications are made in the revised Manuscript (MS) based on the Referees' suggestions and are annotated to ease tracking changes. The Author Comments (ACs) are distinguished by "Italic" typography throughout the text to facilitate the distinction with Referees' Comments (RC1 and RC2).*

**1.1 Referee #1**

**General comments:**

Authors presented results of a numerical study aimed at understanding the impact of including the diurnal variation in prior surface $CO_2$ fluxes, as opposed to using flat daily mean fluxes without diurnal variation. The resulting flux changes in the global and regional models at the level of annual mean fluxes for large regions have been found to be substantial. The study results may point to a useful direction to revising the inverse modeling setups. The results will be useful to $CO_2$ inverse modeling community, as those help understanding the differences between models resolving the diurnal variation in prior fluxes, and those that don't. The manuscript is well written and can be accepted after minor revisions.

*Thank you for concisely outlining the objectives of the study and for the positive feedback!*

**Detailed comments:**

Lines 60-75 Some model related uncertainties in simulating diurnal cycle have been studied by Patra et al. 2008, Law et al. 2008. Could be worth mentioning.

*Thank you for the suggestion! Indeed, these studies are very much in line with our study and have been indicated in the revised MS (L 83-88).*

Line 192 Suggest adding short description of the diurnal flux dataset (from Bodesheim et al) constructed from FLUXCOM), citing time resolution, meteorological field used to drive diurnal cycle of GPP, net annual flux difference between diurnal and daily versions.

*As requested, an expanded description on the flux product used in our study has been added in the revised MS (L 235-249). We would also like to highlight an updated product name: previously we used name 'FLUXCOM' when referring to the flux data product. However, during the review process, the authors have submitted a dedicated manuscript describing it, in which they refer to it as X-BASE (Nelson and Walther et al., 2024, in review). We have therefore updated the naming throughout our MS accordingly.*

Line 275-284 Data in Figure 5 are interesting specifically to $CO_2$ inverse modelers and are worthy of more comments. For example, do corrected budgets increase or reduce mean regional flux dipoles? In addition to temperate North America, emissions grow in tropical South America, and North African sink increased, and 2 later regions are not strongly constrained. Maybe those changes are correlating with differences between models in model ensembles like Friedlingstein et al 2023, Byrne et al. 2023?

*Thank you for the interesting suggestion. We added further explanations regarding the flux impact on dipoles in the results section corresponding to Fig. 5 but also in the discussion relevant to these results in the revised MS (L 347-359). Also, these interpretations are discussed in light of the relevant results reported in literature such as the recent global carbon budget by Friedlingstein et al (2023) and Byrne et al. (2023), which you kindly mentioned. This has expanded the Discussion Section (L 509-521) in the revised MS.*

*In fact, the diurnal effect of $CO_2$ suggests that regional flux dipoles can be modified as there are large differences calculated over tropical and temperate lands, over which dipoles are extreme (please see the attached Figure 1, below). Additionally, the areas with large spatial uncertainty (SD) estimated by Byrne et al (2023) coincide with those having large corrections in our study. In addition to other known reasons such as transport and lack of observations, this indicates that at least part of these uncertainties is likely caused by either missing or inaccurate diurnal variations of $CO_2$ fluxes calculated by bottom-up models. To the extent that FLUXCOM-X has a good*

*representation of $CO_2$ diurnal variations, we believe the use of these flux products can be beneficial for atmospheric inversions. This is supported by the validation analysis carried out in a recent study by Nelson and Walther et al. (2024)([https://doi.org/10.5194/egusphere-2024-165](https://doi.org/10.5194/egusphere-2024-165)). The study used the recently updated version of FLUXCOM (X-BASE). In spite of some shortcomings concerning the seasonal and interannual variability, this product provides good pieces of information regarding the diurnal variations which cannot be constrained by observations, due to the restriction of dataset assimilation to a specific time window.*

[Figure]

*Figure 1: Distributions of corrections added to NEE estimates (y-axis) due to $CO_2$ diurnal effect as compared to the estimated NEE (x-axis) over TransCom land regions.*

*To discuss the correlations over regional estimates derived from all inverse models, this would require accounting for all the prior flux models used in all inversions to get a better perspective on how much discrepancies in diurnal variations contribute to the overall error and flux dipoles. As anthropogenic emissions in CS are not controlled, the effect of their potential inaccuracies cannot be pinned down in the context of this study because potential biases will remain unaffected by the correction of the diurnal effect. We do agree, however, with the overall suggestion for much broader studies in inverse modelling aimed at quantification of the impacts arising from the lack of*

*observations, inaccuracies in emission inventories, diurnal effect (which this study is focusing on), and other factors that affect the estimates of the biosphere sink.*

**Technical corrections:**

Line 26 correct: FLUXCOM

*Thank you for spotting it! It has been corrected in the revised MS.*

Line 132 hyperlink covers only part of the path.

*This has been fixed in revised MS (L 168).*

**References**

Byrne, B. et al.: National CO2 budgets (2015–2020) inferred from atmospheric CO2 observations in support of the global stocktake, Earth Syst. Sci. Data, 15, 963–1004, https://doi.org/10.5194/essd-15-963-2023, 2023.

Friedlingstein, P., et al.: Global Carbon Budget 2023, Earth Syst. Sci. Data, 15, 5301–5369, https://doi.org/10.5194/essd-15-5301-2023, 2023.

Law, R. M., et al. (2008), TransCom model simulations of hourly atmospheric CO2: Experimental overview and diurnal cycle results for 2002, Global Biogeochem. Cycles, 22, GB3009, doi:10.1029/2007GB003050.

Patra, P. K., et al. (2008), TransCom model simulations of hourly atmospheric CO2: Analysis of synoptic-scale variations for the period 2002–2003, Global Biogeochem. Cycles, 22, GB4013, doi:10.1029/2007GB003081.

*Thank you for providing these interesting references, all of which have been included in the revised MS.*

The manuscript "To what extent does CO2 diurnal cycle impact carbon flux estimates in CarboScope?" by Munassar et al. documents a study which relates to the general topic of the impact of the uncertainties in the diurnal cycle of CO2 biogenic fluxes in global and regional atmospheric CO2 inversions.

Here, I insert and resume my access review and maintain my opinion regarding this manuscript, i.e., that it should be withdrawn, and resubmitted with a major revision of its scope. However, the discussion phase gives the opportunity to the authors to answer my concerns and potentially to propose some major revisions to address them.

*We appreciate the efforts of the reviewer and we address all of the listed concerns below, under the respective comments.*

My main concern is that although the general topic of this study is relevant for the inverse modelling community, and although the authors provide a clear analysis, this study focuses on a specific question which can hardly provide general insights for this community.

*Although we know from the previous studies (e.g., Denning et al. 1995, 1996a,b, 1999; Law et al., 2008; Stephens et al., 2007; Patra et al.,2008) that the diurnal variations of both transport and $CO_2$ fluxes are important to account for atmospheric $CO_2$ inversions, the quantitative effect on the estimates of carbon budgets at continental and regional scales found in our study is really intriguing and suggests the need to revise the validity of $CO_2$ diurnal variations in bottom-up models used (as priors) in $CO_2$ atmospheric inversions. The objective of this study is to draw the attention of the community and further motivate the consideration of errors arising from inaccurate representation (or even absence) of $CO_2$ diurnal variations in the prior, which is likely to contribute to persistent discrepancies in flux estimates derived from atmospheric inversions. This issue is manifested by the spread over inversions results reported yearly by the Global Carbon Project (GCP) (Friedlingstein et al., 2022, 2023). For instance, in the tropics alone (30°S-30°N), the diurnal effect resulted in an annual*

*difference of 0.65 Pg C, where inversions suggest that region to be neutral in terms of $CO_2$ flux, albeit with a large uncertainty (Friedlingstein et al., 2022).*

*Of course, we do not conclude that our results provide the ultimate magnitude of the effect, which is subject to the setup of the experiment design, specifically the biosphere model estimating NEE. Nevertheless, the biosphere flux model FLUXCOM-X used in our experimental design is proven to be realistic, especially in resolving for the diurnal variations (detailed description of these products is found in Nelson and Walther et al. (2024), also the extended description in the revised MS (L 235-249) with reference to a previous comment by RC #1).*

As underlined by "in Carboscope" in the title of the manuscript, the specific type of errors examined by this study is inherent to the specific configuration of the CarboScope global inversion system, which:

- uses prior estimates of the Net Ecosystem Exchange (NEE) with flat diurnal cycles

- does not control the diurnal cycle of the NEE

whereas, in general, global and regional inversions (including the CSR system) use prior estimates of the NEE with diurnal cycles and/or control this diurnal cycle.

*We agree that the study is done using CarboScope configuration that did not include the $CO_2$ diurnal cycle in its biosphere priors. Nonetheless, what is more relevant here is the validity of the biogenic flux model used to represent the $CO_2$ diurnal cycle. In the newest simulations of FLUXCOM updated under X-BASE version (Nelson and Walther et al., 2024, in review), the diurnal variations were found to be in good agreement with observations left for validation. Of note, FLUXCOM has not been used so far in global inversions except in this study.*

*Additionally, although as the reviewer has correctly pointed out, most of the global inversions account for diurnal variations in their priors, an important question that need to be raised is: 'to what extent do these variations represent the truth?' According to the data gathered through personal communication, most of groups contributing to GCP include diurnal variations in their bottom-up models through downscaling of weekly/monthly to hourly fluxes using meteorological parameters such as temperature*

*and remotely sensed data, using the approach of Olsen and Randerson (2004). Please find the specific listing for each GCP model in the answer to the next comment.*

*With this in mind, our study thus does not only present a special case concerning CarboScope as the reviewer implies, but in fact assesses a more general issue by analysing the impact of $CO_2$ diurnal cycle derived from the statistical sophisticated biosphere model FLUXCOM-X.*

*We agree that indicating "in CarboScope" in the title gives an impression as the study has an only implication in CarboScope, so "using CarboScope" is more appropriate, therefore we have updated the title in this way.*

Consequently, the manuscript assesses the impact of using and keeping in the global inversions flat diurnal cycles for NEE, and misses broader questions for the inverse modelling community:

- what is the impact of the current level of uncertainties in the diurnal cycles from "bottom-up" products such as simulations from state-of-the-art vegetation models ?

*This is indeed a very important and relevant question. Our study does not depart far from this objective but provides quantitative evidence and draws the attention of the modelling community to consider revising the current state of knowledge regarding the validity of $CO_2$ diurnal cycle calculated via biosphere flux models used as priors in Bayesian frameworks. Although a follow-up study is expected to tackle the issue raised above specifically, it still needs time and efforts to bring all relevant scientific groups together, as the required experiments would need a high amount of coordination to proceed in. Hence, the results we present here serve as basis for the upcoming work but motivate the inverse modelling community to look into this problem more seriously.*

*Through personal communication, we got some details regarding the current state of the diurnal cycle treatment in the global inversions, which contribute to GCP. Here we list brief descriptions of the inversions and the corresponding priors as stated by the providers:*

- ▪ *In CT-NOAA (Carbon Tracker NOAA), the biosphere model computes monthly-average fluxes only, and we use the Olsen & Randerson (2004) method to downscale them to hourly, using ERA5 temperature and radiation.*

- *In CAMS, the prior fluxes are three-hourly from ORCHIDEE (climatological) but the increments are about weekly with night-time and day-time separated.*
- *In CMS-Flux system, we optimize monthly fluxes and specify diurnal cycle in the prior fluxes that were generated following the method proposed by Olsen and Randerson (2004).*
- *In MIROC4-ACTM, the CASA and VISIT monthly-mean fluxes are downscaled to 3-hourly time intervals by redistributing respiration and gross primary production (Olsen and Randerson, 2004) using JRA-55 meteorology, i.e., 2 m air temperature and incoming solar radiation at the earth surface*
- *For NISMON-CO2, VISIT is used at monthly data (Ito, 2019) with 3-hourly downscaling factors applied to GPP and RE, which is already mentioned by others (i.e., Olsen and Randerson, 2004).*
- *For GCASv2, NEE is simulated using the BEPS model, which can simulate hourly NEP.*
- *For GONGGA, the prior fluxes include diurnal cycles, with NEE from ORCHIDEE-MICT 3-hourly simulations.*
- *In IAPCAS, terrestrial biosphere-atmosphere exchange flux includes the diurnal cycle, from 3-hourly CASA v1.0 data (Olsen and Randerson, 2004).*
- *For CTE, we make 3-hourly inputs from hourly original SiB4 fluxes for the biosphere.*
- *In UoE EnKF inversions, we include 3-hourly CASA biosphere fluxes as biosphere priors.*
- *In COLA, daily a priori land (SiB4) fluxes are used, so no diurnal cycle included.*
- *For THU system, 3-hourly fluxes from the hourly SiB4 fluxes are used for the biosphere.*

*As can be seen, most of the inversions use Olsen and Randerson (2004) to include diurnal variations in the prior biosphere models. Unfortunately, none of the listed systems provides a systematic verification and validation in terms of diurnal cycle of the priors being used. Such verification should ideally be performed on a regular basis*

*by comparing the $CO_2$ diurnal cycle to independent observations, such as eddy covariance flux observations across the globe. In this particular aspect, FLUXCOM-X products can be more reliable to represent the diurnal variations based on cross-validation analysis performed by Nelson and Walther et al. (2024, [https://doi.org/10.5194/egusphere-2024-165](https://doi.org/10.5194/egusphere-2024-165)). For a quick look, please see the comparison in the next plot we did using datasets in Nelson et al. (2024), indicating model predictions associated with SD of the differences between model and observations over 29 validation sites distributed around the world. This makes our analysis more robust with the quantitative evaluation of $CO_2$ diurnal effect in inverse modelling.*

[Figure]

*Figure 2: Comparison of predicted NEE with Eddy Covariance observations withheld for validation.*

- what is the capacity of inverse modelling systems to control the NEE diurnal cycle at global and regional scales when only the daytime data are assimilated at most of the measurement stations

*This is in fact a genuine challenge to all the inversions, particularly those assimilating in situ measurements, which is the case in most of the inversions as surface measurements are more accurate to sample $CO_2$ mole fractions within the boundary layer, unlike satellite measurements that retrieve the total column. The issue of not using night time measurements is not only restricted to the representation of measurements but also to the difficulty for transport models to represent the nocturnal boundary layer. These together remain standing challenges to the community and are beyond the scope of our study. Notwithstanding, in Bayesian inversions the diurnal*

*cycle of $CO_2$ can be constrained by the prior biosphere models, as far as such models are capable of capturing the diurnal variations.*

And indeed, the conclusion states (l 464-465) : "Hence, an assessment on the uncertainty of the diurnal cycle effect among atmospheric inversions will be presented in a follow-up study." I think that the two studies should be merged, or at least that the first study should take further steps in the direction of the second. In my opinion, this initial step of assessing the impact of flat diurnal cycles in inversions that do not control this cycle can hardly be the stand-alone subject of a publication in ACP.

*In line with several reasons clarified above in the context of the previous comments, we argue that this study offers lines of evidence with a quantitative assessment regarding the possible order of magnitude of the diurnal cycle effect in estimated $CO_2$ flux budgets. The study also paves the way to the following study and at the same time motivates the community, at least those taking part in GCP, to revise and improve the methodology applied to generate the diurnal cycle of $CO_2$, as well as to stimulate more engagement among the community in the follow-up study.*

This general concern is strengthened by other issues:

- The limitations of the scope in this study are exacerbated when analyzing the results from regional scale inversions by the focus on coupling a global inversion without flux diurnal cycle and a regional inversion with a flux diurnal cycle. This specific configuration corresponds to existing systems and brings insights on the impact of biases in the boundary conditions on regional inversions, but in the spirit of the study and of the introduction of this manuscript (also of the abstract, which is misleading regarding this), one would have also expected the coupling between a global inversion and a regional inversion both without flux diurnal cycle.

*The scope of the study generally focuses on analysing the impact on flux estimates at different spatial scales when the $CO_2$ diurnal cycle is missing in the global inversion. And then as an indirect consequence, the effect passed on to the regional inversions through the boundary condition (normally provided by global inversions) was assessed using CarboScope-Regional that does include diurnal variations in its priors, but also has different setup than CS (please see Table 1). This is clearly indicated all over the*

*manuscript sections. For example, in the introduction L 144-146: "Even though the set-up of the regional inversion does include the diurnal cycle in the a priori fluxes, it is nevertheless prone to biases passed on through the lateral boundary conditions calculated by the global inversion currently not taking the diurnal cycle of $CO_2$ fluxes into account.". In the methods, it is explicitly listed in Table 1 and throughout the text.*

*We admit this piece of information was not mentioned in the abstract, so it has been added in the revised MS (L 23-24).*

- I do not really understand the discussions on the rectifier effets here. Is it useful to have such discussions when dealing with atmospheric inversions relying on dynamical models which account for the variations in the vertical mixing (even if with some limited accuracy) ? In a more general way, is not the introduction going back too far ? Several parts of these discussions are quite difficult to follow (in the introduction and in the conclusion) and sometimes misleading, with lines 71-72 stating that "CO2 concentrations are lower near the surface than in the free atmosphere due to strong daytime vertical mixing" while the daytime vertical mixing attenuates the decrease of CO2 near the surface due to the photosynthesis.

*Even though it is worth mentioning the rectifier effect in the introduction as part of the research background, we agree that this study does not deal with the rectifier effect but with the diurnal cycle effect of $CO_2$ in simulating hourly mole fractions. So, we modified and rephrased the relevant paragraphs in the introduction accordingly (L 89-90 and L 94-95).*

- The manuscript assumes that the linearity of the impact of ignoring the diurnal cycle in both the prior estimate of the fluxes and the inversion control vector is obvious. However, it is due to the configuration of the CS system (which would add its correction for the daily fluxes to the prior hourly fluxes as a constant value over the day rather than scale the prior hourly fluxes) and in practice, the variational inverse modelling scheme loses part of this linearity. Therefore, this topic deserves some explanations.

*In terms of the system configuration, CS does optimize fluxes as an additive correction made to the a-priori fluxes at daily timesteps. Thus, in the current setup neither the a-priori fluxes nor the flux adjustments consist of the diurnal variations. So, instead of performing inversion that adds corrections to the hourly prior fluxes, we inverted the*

*differences in mole fractions (simulated with daily and hourly fluxes) that reflect the variability needed to be added to the daily mean fluxes so that a diurnal cycle is reconstructed. This is because the inversion operator is fully linear since daily a posteriori fluxes, which would contain diurnal variations, and daily a posteriori fluxes optimized without diurnal variations only differ in the variability around the daily mean of the a-priori fluxes, i.e., (dx–0), where dx here refers to hourly variability. This means, anything else will cancel out when subtracting such two inversions. And then these inverted differences represent the diurnal-effect corrections added to the CS posterior estimates, which lack the diurnal cycle of $CO_2$.*

*Further explanations in accordance with the suggestions have been included in the relevant text in the revised MS (L 184-186 and L 256-261).*

- The manuscript could have expanded the discussions on the signal from the fluxes at the observation sites (depending on the type of observation site and on the periods of the day when their observations are assimilated) that is exploited by the inversions: an integration in time and space of signal or a time-lagged signal from remote fluxes vs. the differences between stations informing about the fluxes in between vs. instant signal from local fluxes; this may help better understand the positive and negative biases in the observations and flux estimates depending on the stations and regions. Why would some of the sites "affected by large-scale ocean background" correspond to large negative biases in the TM3 usual simulations if they bear little terrestrial influences as currently stated by the first paragraph of section 3.1 (the following paragraphs provide a quite different picture, but ignore the large negative biases) ? This section forgets to discriminate results depending on whether the observations are assimilated during nighttime or daytime only.

*Thank you for the suggestion. We have expanded the discussions on the results reported in Sect. 3.1 according to the suggestion in the revised MS (L 271-273, L 276-278, L 281, and L 285-291).*

*Indeed, the distinction between negative and positive differences in this respect can be deduced from the type of sites that reflect the representativeness of their backgrounds. This is obviously shown in Fig. 2 left panel where the differences are clustered depending on the location site category. The most dominant positive*

*differences are seen over large number of stations that sample the biosphere signal during day-time like towers, continental, and surface sites and therefore we see an overestimation of $CO_2$ sources if daily means of NEE are considered. This implies that simulations miss the uptake flux signal during the day. This impact is dominating the inversion results (also in our analysis) as most of the stations (96 sites) represent day-time terrestrial signal in which best vertical mixing conditions usually occur. When looking at the 29 remote sites, the differences due to the diurnal effect are almost around zero (0.02 ppm), compared with about 0.75 ppm resulting from the first suite of sites. This is actually reasonable because over such sites ocean background dominates and thus does not vary much with daily and hourly NEE. Similarly, with the simulations calculated over locations with no (or weak) biosphere signal like ocean and aircraft sampling locations where the land signal is actually lagged (which interprets the flipped sign) compared to sampling instant flux signals at terrestrial land locations. For the mountain sites, the mean difference calculated over 13 sites shows a negative value of -0.22 ppm. Here we emphasize that simulations are confined to a night-time window, which explains the negative sign between daily- and hourly-NEE-based simulations as $CO_2$ accumulates near the surface due to respiration but also contributions from the residual layer forming in the free troposphere.*

That said, as indicated earlier, I think that the authors conduct a clear and sensible analysis. Pieces of information are missing in the presentation of the inversion configuration (e.g., regarding the prior error covariances of CSR). Furthermore section 3 is sometimes a bit confusing regarding the sign of the biases since it discusses both the differences inversion with diurnal cycle minus inversion without diurnal cycle and the biases, which correspond to the inversion without diurnal cycle minus the inversion with diurnal cycle. This culminates at lines 307-308 where the underestimation (which should correspond to the bias in the usual CS-CSR inversions) of the CO2 uptake if found during the growing season. The discussions on the IAV may also be led a bit too fast. However, overall, the paper reads well.

*We thank the reviewer for this remark and we have updated the text accordingly. Description on CSR configurations regarding the prior error covariances has been added in the revised MS (L 188-191).*

*Regarding the sign of biases, this has been addressed in a previous comment, so we kindly refer you to it. For a quick answer, what has been done in our analysis is starting with daily- minus hourly-NEE-based simulations, so the inverted difference between a daily posterior without diurnal cycle and a daily posterior with diurnal cycle would require adding an annual correction in the magnitude of this difference, regardless of its sign and season. The opposite holds true when starting with hourly- minus daily-NEE-based simulations. Modifications have been made in the revised MS accordingly (L 365-366, L 388-392).*

*Additionally, the discussion on the IAV is further expanded in the revised MS. We clarified the possible impact of site network and coverage of time on IAV, which should be taken into account as the diurnal effect can result in a different response, should the site network be changed (L 527-530).*

---

## Author Response (AR2)

**Point-by-Point response to RC#2**

*Again, we thank the two anonymous reviewers as well as the handling Editor for their time to go through the revised manuscript and for the feedback on our response to the comments raised in their reviews.*
*In the following text, we address point-by-point the remaining comments raised by the second reviewer (RC#2). Note: Author Comments (ACs) are formatted in "Italic" indented text in correspondence to each RC.*

I thank the authors of " To what extent does CO2 diurnal cycle impact carbon flux estimates using CarboScope?" for their detailed answers to my comments and for their revision of the manuscript.

> *Thank you for going through the answers and the manuscript.*

These answers bring some arguments in favor of publishing this study. I do not find them totally convincing but their are significant enough to open the perspective of reaching a suitable balance.

> *The remaining concerns are addressed with additional analysis/modifications included in the revised MS.*

The authors show that many of the GCP global inversions use the approach of Olsen and Randerson 2004, and they implicitly assume that such an approach poorly catches the actual NEE diurnal cycles.

> *This is addressed in the context of the next comment.*

Therefore, a more convincing demonstration of the need to "revise and improve the methodology applied to generate the diurnal cycle of CO2", a better way to "draw the attention of the modelling community to consider revising the current state of knowledge regarding the validity of CO2 diurnal cycle calculated via biosphere flux models used as priors in Bayesian frameworks" could be based on the estimate of the changes in flux inversions when using this Olsen and Randerson 2004 approach vs. FLUXCOM-X to define the diurnal cycle of the prior NEE (in addition to the current test with flat vs. FLUXCOM diurnal cycles).

> *Thank you for the suggestion! This analysis has been added to the discussion (L 448-468). We added results from CASA-based biogenic fluxes that were downscaled to hourly NEE by Carbon Tracker (CT– NOAA) based on the method presented by Olson and Randerson (2004), where it is typically used in CT global inversion.*

The authors could, at least:

1) better document, in the introduction, the dominant types of prior estimates of the NEE diurnal cycles in current global and regional inversions. I find the sentence "However, not all atmospheric inversions account for the effect of diurnal cycle in biosphere-atmosphere exchange of CO2" (l. 92) quite misleading regarding this.

*We added it to the introduction in the revised MS (L 92-109).*

2) demonstrate, using past publications, that the level of the uncertainty (including biases) arising from methods like that of Olsen and Randerson 2004 and even from vegetation models with sub-diurnal resolution is so large that the order of magnitude and general mapping of the corresponding uncertainty in the flux inversions can be assessed from experiments with a flat diurnal cycle.

*As a first indication, the additional analysis included in the discussion (based on the suggestion above) shows that there are large discrepancies in the diurnal variations between FLUXCOM-X and one product of Olson-Randerson-based methodology. We actually took into account comparing results available from other studies in the discussions. However, given the lack of studies regarding the diurnal cycle uncertainty among biosphere models, or methodologies applied to generate the diurnal cycle, it is still an open issue that calls for a dedicated study to investigate the level of uncertainty amid the current biosphere models, at least those used as priors in inversions. This is actually the objective of the follow-up study, which requires broader collaborations among the modellers.*

The general discussion in our sequence of review / response regarding the relevance of these experiments should be better reflected in the manuscript.

*All the points raised during the rounds of revisions are included.*

Other points:

- title: I suggest generalizing "using CarboScope" to something like "global and regional atmospheric inversions", making sure that the manuscript definitely supports such a generalization (see above). Similar comment for line 95: the general objective of the publication should be broader.

*Thank you for the suggestion! The title is now slightly modified based on the suggestion: "To what extent does $CO_2$ diurnal cycle impact flux estimates derived from global and regional inversions". Also, the objective indicated in the introduction has changed accordingly in the revised MS (L 110).*

- I am still concerned by the following sequence of sentences (lines 71-75): "That is, in the early afternoon on any sunny day of the growing season, atmospheric CO2 concentrations are lower near the surface due to active photosynthesis while CO2 is lifted up and diluted by the strong daytime vertical mixing (Stephens et al., 2007). By contrast, during nighttime CO2

concentrations accumulate near the Earth's surface owing to ecosystem respiration under a shallow boundary layer. On average, the covariations between the atmospheric transport and terrestrial biospheric fluxes... »

=> I may misunderstand the term " covariations " here. At night, both the emissions and the lack of vertical mixing tend to increase the level of the concentrations close to the surface. But, again, during daytime, the high vertical mixing tend to attenuate the decrease of the concentrations close to the surface due to the photosynthesis, i.e. the overall diurnal cycle of concentrations close to the surface. So what are these " covariations " about ? The answer to this question may push for a simplification of this section of the introduction (see my comment in the previous review regarding this) and for a better focus of this introduction on current practices in atmospheric inversions (see above).

> *According to the suggestion, we rephrased this section in the revised MS (L 70-75), so as to clarify the concept of covariance between atmospheric vertical mixing and terrestrial fluxes. "Covariations" was basically referring to the covariance between the vertical transport and biogenic fluxes due to the same forcing initiating both.*